# Using modelled discharge to develop satellite-based river gauging: a case study for the Amazon Basin

Jiawei Hou[1], Albert I.J.M. van Dijk[1], Luigi J. Renzullo[1], Robert A. Vertessy[1, 2]

[1]Fenner School of Environment and Society, Australian National University, Canberra, Australian Capital Territory, Australia
[2]School of Engineering, University of Melbourne, Melbourne, Victoria, Australia

*Correspondence to*: Jiawei Hou (jiawei.hou@anu.edu.au)

**Abstract.** River discharge measurements have proven invaluable to monitor the global water cycle, assess flood risk, and guide water resource management. However, there is a delay, and ongoing decline, in the availability of gauging data and stations are highly unevenly distributed globally. While not a substitute for river discharge measurement, remote sensing is a cost-effective technology to acquire information on river dynamics in situations where ground-based measurements are unavailable. The general approach has been to relate satellite observation to discharge measured in situ, which prevents its use for ungauged rivers. Alternatively, hydrological models are now available that can be used to estimate river discharge globally. While subject to greater errors and biases than measurements, model estimates of river discharge do expand the options for applying satellite-based discharge monitoring in ungauged rivers. Our aim was to test whether satellite gauging reaches (SGRs), similar to virtual stations in satellite altimetry, can be constructed based on MODIS optical or GFDS passive microwave derived surface water extent fraction and simulated discharge from the World-Wide Water (W3) model version 2. We designed and tested two methods to develop SGRs across the Amazon Basin and found that the Optimal Grid Cell Selection method performed best for relating MODIS and GFDS water extent to simulated discharge. The number of potential river reaches to develop SGRs increases from upstream to downstream as rivers widen. MODIS SGRs are feasible for more river reaches than GFDS SGRs due to its higher spatial resolution. However, where they could be constructed, GFDS SGRs predicted discharge more accurately as observations were less affected by cloud and vegetation. We conclude that SGRs are suitable for automated large-scale application and offer a possibility to predict river discharge variations from satellite observations alone, for both gauged and ungauged rivers.

## 1 Introduction

River discharge data are used to monitor the global water cycle, assess flood risk, and guide water resource management (Brakenridge et al., 2012). Example applications also include: assessing the contribution of river flow to oceans and the distribution of river runoff on continents; training models to predict how water resources will be affected under climate change; identifying where flood intensity and frequency is likely to increase; providing information for flood forecasting,

monitoring and warning systems; and better formulating water allocation plans for domestic, agricultural, and industrial uses (Van Dijk, 2015).

Over the past century, many ground-based gauging stations have been built to monitor river discharge across the world (Dai et al., 2009). However, the number of accessible gauging station records has decreased over the years due to the reluctance of contributors to share data, or the lack of financial and technical support to maintain gauging stations (Vörösmarty, 2001; Biancamaria et al., 2011; Brakenridge et al., 2012; Fekete et al., 2012). In addition, gauging station networks are sparse and unevenly distributed. For instance, there are few gauging stations on rivers with braided channels or wide floodplains, and on rivers located in remote areas (Smith et al., 1996; Alsdorf et al., 2003; LeFavour and Alsdorf, 2005; Calmant and Seyler, 2006). Finally, gauging stations are only representative for a single point along a river, which can make it difficult to obtain insight into hydrological conditions throughout river networks (Hunger and Döll, 2008; Stahl et al., 2012).

Remote sensing is a cost-effective way to acquire information on river dynamics both at regional and global scales (Alsdorf et al., 2007). Satellite observations can cover a river in the lateral dimension where there are wide channels or broad floodplains and in the longitudinal dimension in long and complicated river systems (Smith, 1997; Bjerklie et al., 2003). Whereas gauging stations measure water level, remote sensing typically measures river extent or width with the exception of river altimetry (Birkett et al., 2002; Coe and Birkett, 2004; Kouraev et al., 2004; Zakharova et al., 2006; Papa et al., 2010). Such satellite-based measurements can be related to measured river discharges. The general approach has been to develop rating curves relating satellite observation where they coincide with in situ river discharge measurement, and to use the fitted rating curves to estimate river discharges with satellite observations only (e.g. Revilla-Romero et al., 2014).

Optical and microwave satellite imaging can provide continuous spatial observations of surface water extent along the entire river channel. Both inundation-discharge and width-discharge relationships can be developed using ground measurements of river discharge and satellite optical or synthetic aperture radar (SAR) imagery (Smith et al., 1995; Smith et al., 1996; Papa et al., 2008; Smith and Pavelsky, 2008; Pavelsky, 2014). In addition, Brakenridge et al. (2007), Tarpanelli et al. (2013) and Van Dijk et al. (2016) demonstrated that the ratio of a calibration and measurement pixel remote sensing signal for MODIS near infrared reflectance or AMSR-E passive microwave brightness temperature can provide an indicator of variations of river discharge, which provides opportunities to monitor river discharge at a global scale with medium spatial resolution and high temporal resolution. However, optical remote sensing requires a clear view of the water surface, unobscured by cloud or a dense vegetation canopy. While radar and passive microwave remote sensing are not affected by these factors to the same extent, radar is susceptible to wind-induced waves and vegetation above surface water, whereas the resolution of passive microwave imagery is too coarse for many rivers. As an alternative to the rating curve approach, open-channel hydraulic equations such as the Manning equation can be used to estimate river discharge from remotely sensed data. However, in addition to remotely sensed data, additional field data including river depth and roughness coefficient are needed to apply

this method and can introduce large uncertainties, which limits its predictive performance (Te Chow, 1959; LeFavour and Alsdorf, 2005; Jung et al., 2010; Woldemichael et al., 2010; Michailovsky et al., 2012).

The main disadvantage of all methods described above is that in situ measurements are still necessary, which makes it impossible to apply them at ungauged sites and unsuitable for automated large-scale applications. An alternative is to use hydrological models to estimate river discharge throughout river networks and to relate these estimates to satellite imagery. In this paper we investigate whether satellite gauging reaches (SGRs) can be established at both gauged and ungauged rivers and applied to provide continuous, consistent, and up-to-date river discharge monitoring over a large area. An SGR, by analogue of an in situ gauging station, is constructed based on an automated statistical method which relates hydrological model simulated river discharge to optical or passive microwave-derived surface water extent fraction for a region that includes the river reach. The concept of a SGR is similar to that of a 'virtual station' used in satellite altimetry (Calmant and Seyler, 2006), but acknowledges that river surface water extent is measured along a river reach rather than at a single location. In the first part of this paper, we design and compare two methods to construct SGRs, and then choose the best method and evaluate its performance. In the second part, we construct SGRs based on optical or passive microwave observations and simulated river discharges, then compare river discharge estimates from optical and passive microwave observations and from the hydrological model against in situ river discharge measurements. We hypothesize that SGRs may perform better than the hydrological model if the model has poor timing; or worse if the model is already quite good. In the latter case, however, SGRs may still be useful for monitoring river discharge in the absence of a real time hydrological model or gauging stations.

## 2 Data and methods

The fundamental assumption in our methodology is that there exist strong, monotonic relationships between remote sensing signal, surface water extent, river channel storage, and river discharge. Surface water extent fraction (hereafter, *water extent*) was previously derived from GFDS passive microwave and MODIS optical remote sensing signal by Van Dijk et al. (2016). River storage and discharge were estimated by the World-Wide Water (W3) model version 2 (Van Dijk et al., 2018). First, we designed two alternative methods to develop SGRs with the aid of hydrological model estimates and compared performance of these methods on rivers of different sizes. We then applied the method that performed best across the Amazon Basin. Second, SGRs were constructed across the Amazon Basin based on MODIS and GFDS water extent. The derived river discharge estimates from the SGRs and from the W3 model were evaluated against in situ river discharge measurements at 31 stations. The overall methodology is shown in Figure 1.

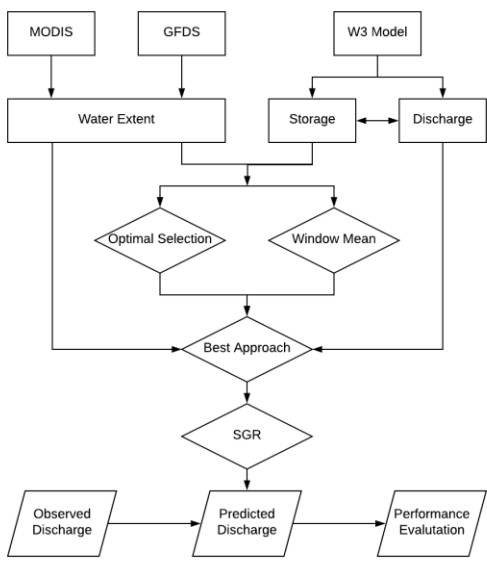

**Figure 1** Workflow of the overall methodology (rectangle: data; diamond: method; parallelogram: validation).

## 2.1 Study region

We chose the Amazon Basin as a case study in this research. The Amazon Basin serves as a suitable testbed for our method in that it contains numerous inaccessible river reaches surrounded by dense tropical rainforests, frequently flooding areas, extremely wide river floodplains, and braided river channels. Moreover, it has unregulated rivers of widely varying size, which provides an opportunity to assess the sensitivity of spatial resolution in remote sensing to river size. In addition, because rainfall estimates across the Amazon Basin are generally poor, it is meaningful to test whether modelled discharge can be improved through remote sensing. A challenge is that MODIS observations are often affected by cloud cover. Van Dijk et al. (2016) found strong correlations between optical and passive microwave-derived water extent estimates and station discharge observations in the Amazon Basin, from which we infer that there may be further opportunities to develop satellite-based river gauging using modelled discharge at ungauged sites.

## 2.2 Data

### 2.2.1 Remote sensing

The Global Flood Detection System (GFDS) was developed to monitor floods and is operated by the Joint Research Centre of the European Commission, in collaboration with the Dartmouth Flood Observatory. De Groeve et al. (2015) proposed a discharge signal, *s,* as the ratio of brightness temperatures between a targeted wet pixel (measurement pixel) and a nearby dry pixel (calibration pixel), which allows tracking at relative changes in surface water extent within a river reach. The discharge signal *s* was calculated from brightness temperature recorded at 36.5 GHz in the H-polarization by the Japanese Space Agency's AMSR2 and TRMM TMI sensors and NASA's AMSR-E and GPM instruments. The GFDS raster data

product used here, named 'merged 4-day average datasets', provides daily $s$ as an average value of the signal for the current day and the signal from the last 3 days, with a spatial resolution of 0.09°×0.1° over the period of 2000-2014.

The Moderate Resolution Imaging Spectroradiometer (MODIS) is an optical sensor aboard the NASA's Terra and Aqua satellites, which provide two images per day for almost every point on the planet. The surface observing capability of MODIS is limited by cloud cover, but this can be mitigated by using MODIS 8-day or 16-day composites which reduce the influence of cloud contamination. The MODIS data used here is the shortwave infrared (SWIR) spectral band 7 (2105–2155 nm) data from the MCD43C4.005 product which contains 8-day Nadir BRDF (Bi-directional Reflectance Distribution Function) adjusted reflectance (NBAR) composites of imagery over the period of 2000-2014. The optical data was aggregated to a spatial resolution of 0.05 °×0.05 ° for rapid processing. The method to calculate surface water extent fraction from GFDS and MODIS data was described by Van Dijk et al. (2016). We calculated both 8-day and monthly GFDS and MODIS-derived surface water extent fraction across the Amazon Basin.

### 2.2.2 Hydrological model

The World-Wide Water (W3) model version 2 (Van Dijk et al., 2018) is a global implementation of the Australian AWRA-L model; a grid-based, one-dimensional water balance model with semi-distributed representation simulating soil, groundwater and surface water stores (Van Dijk, 2010). AWRA-L is used operationally by the Australian Bureau of Meteorology to estimate daily water balance component across Australia at a spatial resolution of 0.05 °×0.05 ° (Frost et al., 2016). Each grid cell has three soil layers (top, shallow and deep soil layers) and one unconfined groundwater layer, and hydrological processes considered in the model include: (1) net precipitation and interception losses; (2) saturation excess overland flow, infiltration excess surface runoff, and infiltration; (3) soil water evaporation, drainage and interflow; (4) groundwater evaporation and base flow; (5) vegetation transpiration and cover adjustment; (6) surface water evaporation, inflows from runoff and discharge, and catchment water yield. Details about the W3 model including input data, parameterization, calibration and validation can be found in Van Dijk et al. (2018). This model was not calibrated against gauging data used in this study. Daily simulated river channel storage and discharge in 0.05 °×0.05 ° grid cells were used in this research and averaged to 8 days to relate them to remote sensing data. The W3 model estimates of river channel storage, rather than discharge, are compared with optical and passive microwave-derived water extents, because conceptually they are more closely related. However, river channel storage has a linear relationship with discharge within the W3 model structure.

### 2.2.3 In situ river discharge measurement

Monthly in situ river discharge measurements were collected from two datasets developed by Beck et al. (2015) and Dai (2016) respectively. The former dataset was established to combine global unregulated river discharge data from the Global Runoff Data Centre (GRDC) and the USGS GAGES II (Geospatial Attributes of Gauges for Evaluating Streamflow) databases. The same data were used in a precursor to this study (Van Dijk et al., 2016). The latter dataset was developed to

compile river flow data from the farthest downstream gauging stations of the world's largest 925 rivers. Among these two datasets there are 31 gauging stations located inside the Amazon Basin with records that were fully or partially overlapping with the remote sensing and model simulation records.

## 2.3 Method

### 2.3.1 Satellite gauging reach designs and performance evaluations

In developing SGRs, we tested two alternative methods to correlate remotely sensed water extent with modelled river channel storage. Method A finds the most strongly correlated water extent over a search window, which we refer to here as *Optimal Grid Cell Selection*. Method B calculates the spatial average water extent within a search window, referred to here as the *Window Mean*. We experimented with different window sizes: $0.15°\times0.15°$, $0.35°\times0.35°$, $0.55°\times0.55°$, $0.75°\times0.75°$, and $0.95°\times0.95°$ (Table 1). These 10 experiments (two methods for each of the five search windows) were applied for each grid cell of the W3 model along the river channel across the Amazon Basin, using 8-day MODIS and GFDS-derived water extent estimates, respectively. For each grid cell, the steps are as follows: A search window centres on a target grid cell of the W3 model, and simulated storage time series for the target cell and all water extent time series located within the search window are selected. Next, in method A, the storage time series is compared with each water extent time series, and the one with the strongest correlation is chosen to develop the SGR. In method B, spatial average water extent time series across the window is calculated and used to develop the SGR.

**Table 1** Experiment design (window size) for two methods to develop SGRs

| Experiments | I | II | III | IV | V |
|---|---|---|---|---|---|
| Optimal Selection | 0.15° | 0.35° | 0.55° | 0.75° | 0.95° |
| Window Mean | 0.15° | 0.35° | 0.55° | 0.75° | 0.95° |

To test which of the two methods best estimates storage for different river sizes, we divided river reaches into four categories based on their mean simulated discharge over the period of 2000-2014. The four categories of river were defined as small ($10^2$-$10^3$ m$^3$s$^{-1}$), medium ($10^3$-$10^4$ m$^3$s$^{-1}$), large ($10^4$-$10^5$ m$^3$s$^{-1}$) and very large ($>10^5$ m$^3$s$^{-1}$) rivers (Figure 2). We did not consider rivers where discharge is less than $10^2$ m$^3$s$^{-1}$ as we assume that they would have channel widths that could not be resolved using our sensing and modelling methods. The most suitable window overall, and SGR selection method, were subsequently decided upon based on performance statistics.

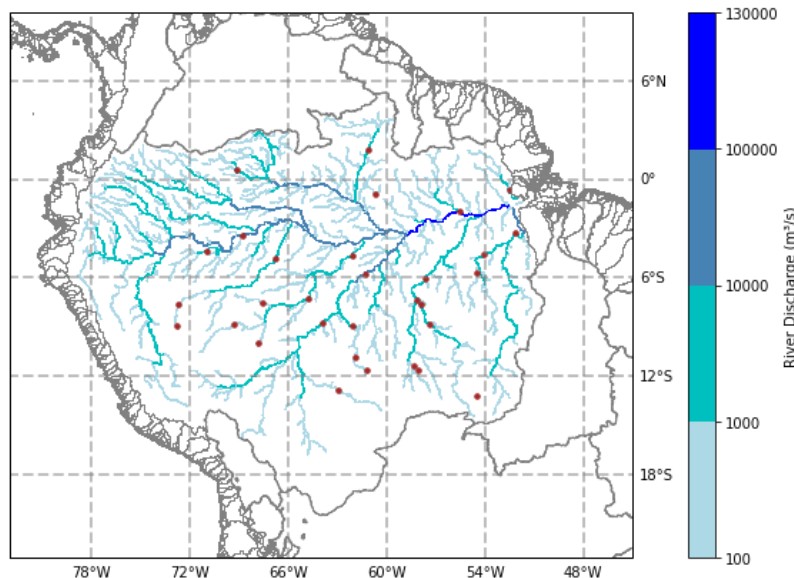

**Figure 2** The W3 model simulated mean river discharges (> 100 m³s⁻¹) in the Amazon Basin (grey line: basin boundary; brown dot: gauging station).

The superior method was applied to construct SGRs across the Amazon Basin, using 8-day MODIS and GFDS water extent, respectively. For method A, the time series was split into training and validation periods to ensure independent validation. Data for the training period were used to select the best correlating pixel for each model grid cell, while data from the validation period were used to evaluate SGRs performance. We evaluated the results from 3 experiments: (I) training: 2005-2014; validation: 2000-2004, (II) training: 2000-2004 and 2010-2014; validation: 2005-2009, and (III) training: 2000-2009; validation: 2010-2014 (Table 2). The mean result was adapted as the overall evaluation statistic. For method B, spatial average water extent for the whole period of 2000-2014 was compared to storage directly, as this produces the same results as using the cross-validation method. The performance of SGRs was assessed using Spearman's rank correlation ($\rho$), since the relationship between water extent and storage is often non-linear.

**Table 2** Training and validation periods for cross-validation method

| Periods | I | II | III |
|---|---|---|---|
| Training Period | 2005-2014 | 2000-2004&2010-2014 | 2000-2009 |
| Validation Period | 2000-2004 | 2005-2009 | 2010-2014 |

### 2.3.2 Evaluations of satellite gauging reaches and the w3 model

A Spearman correlation $\rho > 0.6$ in a grid cell (0.05 °×0.05 °) was used to identify a potential river reach for developing an SGR. We constructed an SGR for this river reach based on water extent and modelled discharge. The developed SGR was

used to estimate river discharges using satellite observations only. We used the same training and validation periods described in Section 2.3.1 (Table 2). In the training period, both model and remote sensing data were used to establish a relationship between water extent and discharge. Remote sensing data for the validation period were used to estimate river discharge from SGRs using the developed relationship. To ensure the relationship can be transferred from model simulation to SGR it was necessary to eliminate systematic differences between the two time-series. Because the distribution of discharge is non-Gaussian, a simple transform by the first two statistical moments produced poor results. Better results were achieved through cumulative distribution function (CDF) matching. Following the approach of Van Dijk et al. (2016), we used a rank-based look-up-table approach to estimate river discharge from mapped estimates of water extent. Estimates of water extent in the validation period are ranked relative to the estimate water extents in the training period, and cumulative distribution function (CDF) matching is then used to provide corresponding river discharge estimates over the validation period. The combination of river discharge estimates from the three validation periods was lumped to represent performance over the whole study period of 2000-2014. Overall, we obtained three river discharge estimates from MODIS, GFDS, and the model. All were then validated and evaluated against monthly in situ river discharge measurement (daily in situ data was not available).

## 3. Results

### 3.1 Evaluations of satellite gauging reach designs

The 10 experiments described in Section 2.3.1 for relating remotely sensed water extent to simulated river channel storage were compared, using MODIS and GFDS water extent, respectively. For MODIS, irrespective of window size or SGR selection method, the mean $\rho$ between water extent and storage increases and the range of $\rho$ narrows as discharge becomes larger (Figure3). For the small rivers ($10^2$-$10^3$ m$^3$/s), the *Optimal Selection* method (method A) achieved mean $\rho < 0.6$, while the *Window Mean* method (method B) resulted in mean $\rho < 0.3$. In contrast, in the main Amazon River channel, method A produced mean $\rho > 0.7$, while method B resulted in mean $\rho > 0.5$. Across all categories of discharge (Figure 3a – d), method A produced $\rho$ that increases as the window size increases, and method B produced inconsistent results. In the same way, the mean $\rho$ in GFDS cases also increases as discharge rises (Figure4). Both methods showed similar results as the mean $\rho$ grows as the window size becomes larger (Figure 4a – d). For small rivers, both methods produced mean $\rho < 0.5$, while they achieved mean $\rho > 0.4$ in the main Amazon River channel. Overall, MODIS performed better than GFDS, and method A performed better than method B. Although the 0.95°×0.95° window size produced better results, larger windows increased the risk of selecting pixels over nearby rivers rather than the target river. We found that using method A with a search window of 0.55°×0.55° was the best overall approach for developing satellite-based river gauging.

This approach was applied across the Amazon Basin using MODIS and GFDS water extent respectively (Figure 5a – b). For MODIS SGRs, there were strong relationships ($\rho > 0.6$) between water extent and storage in most reaches of the main river

channel and its large tributaries, particularly in the larger channels ($\rho > 0.8$), while there were weak correlations ($\rho < 0.4$) in upstream tributaries. The overall performance of the MODIS SGRs was superior to the GFDS SGRs. For GFDS SGRs, there were more river reaches with low correlations ($\rho < 0.4$) in upstream tributaries, and the lower reach of the Amazon River did not show continuous high correlations ($\rho > 0.8$).

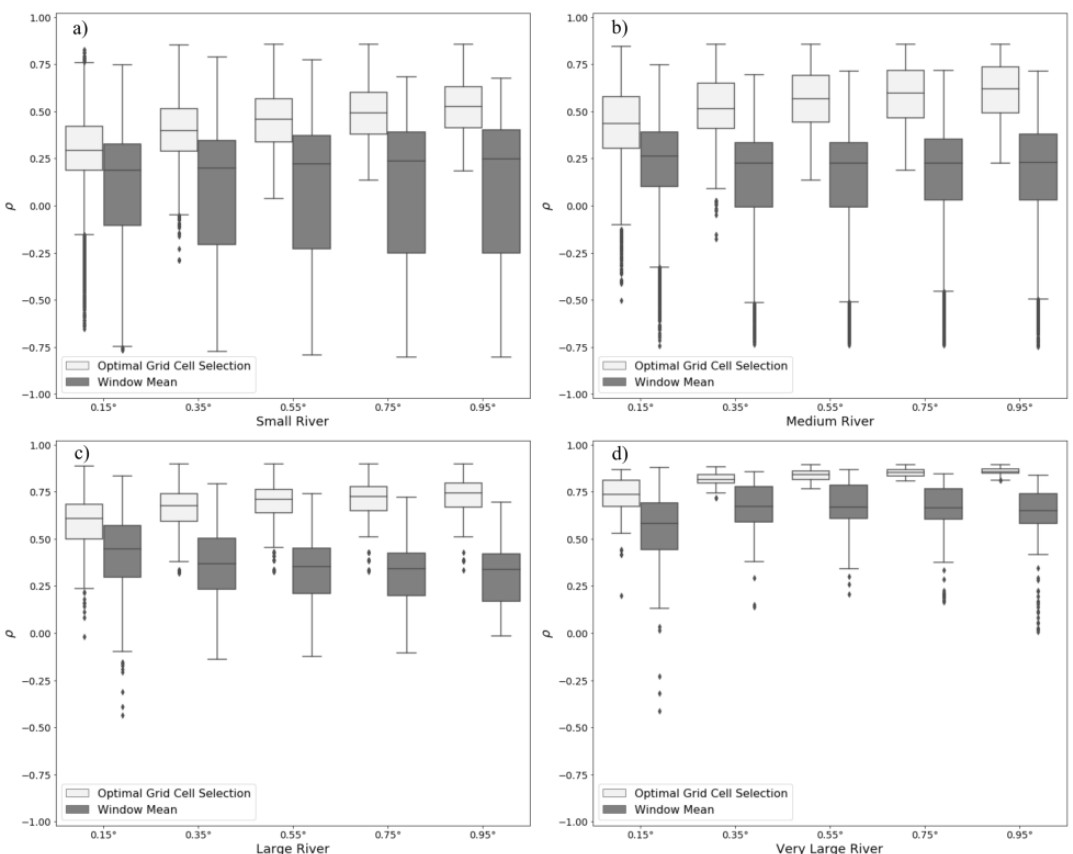

**Figure 3** Distributions of Spearman's rank correlation between MODIS water extent and simulated storage using different window sizes (0.15°×0.15°, 0.35°×0.35°, 0.55°×0.55°, 0.75°×0.75°, and 0.95°×0.95°) and two approaches (light grey: *Optimal Grid Cell Selection* (Method A); dark grey: *Window Mean* (Method B)) in four categories of river flow across the Amazon Basin. Outliers are data beyond the distance larger than 1.5 times the interquartile range from the first and third quartiles.

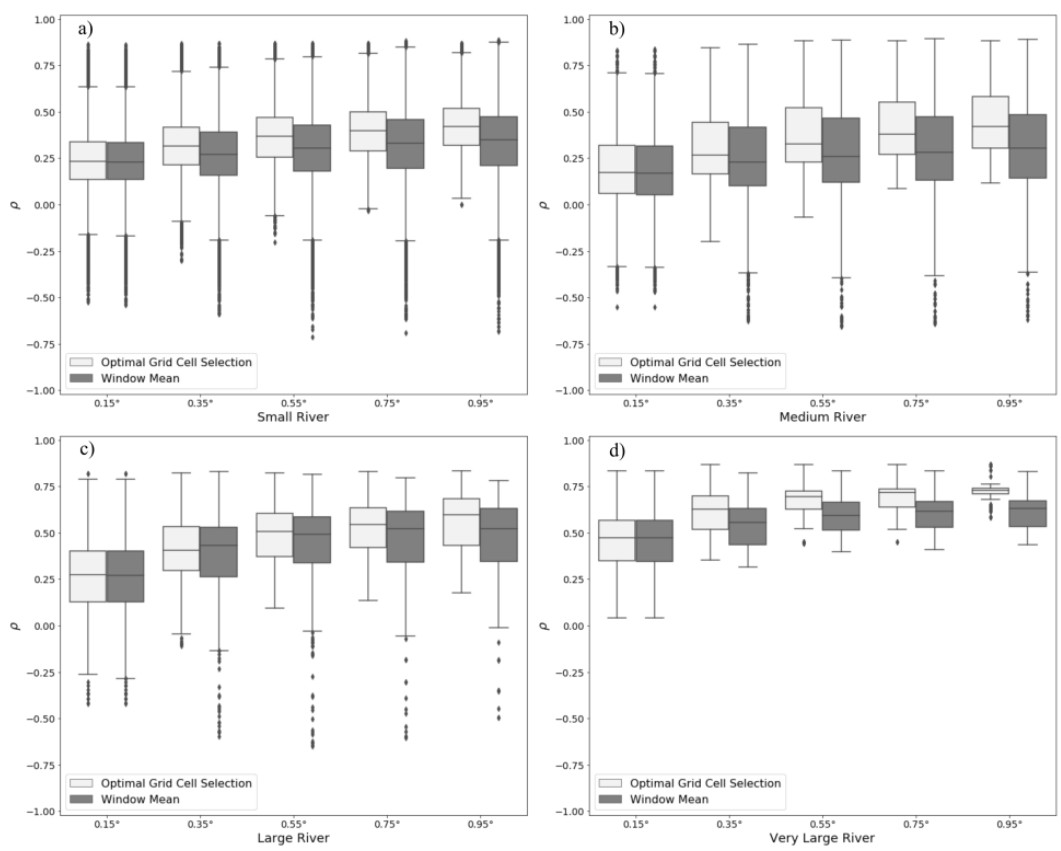

**Figure 4** Distributions of Spearman's rank correlation between GFDS water extent and simulated storage using different window sizes (0.15°×0.15°, 0.35°×0.35°, 0.55°×0.55°, 0.75°×0.75°, and 0.95°×0.95°) and two approaches (light grey: *Optimal Grid Cell Selection* (Method A); dark grey: *Window Mean* (Method B)) in four categories of river flow across the Amazon Basin. Box plots are defined as in Fig.3.

## 3.2 Performance of satellite gauging reaches and the w3 model

We defined river reaches where $\rho$ between water extent and storage is greater than 0.6 as potential locations for developing useful SGRs (Figure 5). While there were 31 gauging stations in the Amazon Basin, only 10 gauging stations coincided with MODIS potential SGR sites and 5 with GFDS sites. Thus, we only assessed river discharge estimates for these 15 cases. Monthly river discharge estimates from MODIS, GFDS and the model for the period of 2000-2014 were compared against monthly in situ river discharge measurements (Figure 6). We focused on flow pattern comparisons between predicted and observed discharges, so different vertical axes were chosen to bring them close to each other (observations from gauging stations are shown on the right axis and river discharge estimates derived using remote sensing and model on the left axis). The W3 model yielded good estimates, with Pearson correlation (*R*) generally greater than 0.8 across most sites. Seven of the 10 MODIS SGRs estimated river discharge with *R* above 0.7, and the SGR for gauging stations G12 and G31 performed best, with *R* close to 0.9. Overall, MODIS SGRs estimates were not as skilful as the model, with the exception of the one for

gauging station G12. While there were fewer potential sites for GFDS SGRs, they were similarly or more skilful than the MODIS SGRs. For gauging stations G12 and G19, GFDS produced stronger river discharge estimates than either MODIS or the W3 model. In total, estimated river discharges from the SGRs and the model showed similar flow fluctuations against in situ river discharge observations. The performance of daily, 8-day and monthly MODIS and GFDS SGRs are compared and

5    discussed in the Supplement (Figure S1).

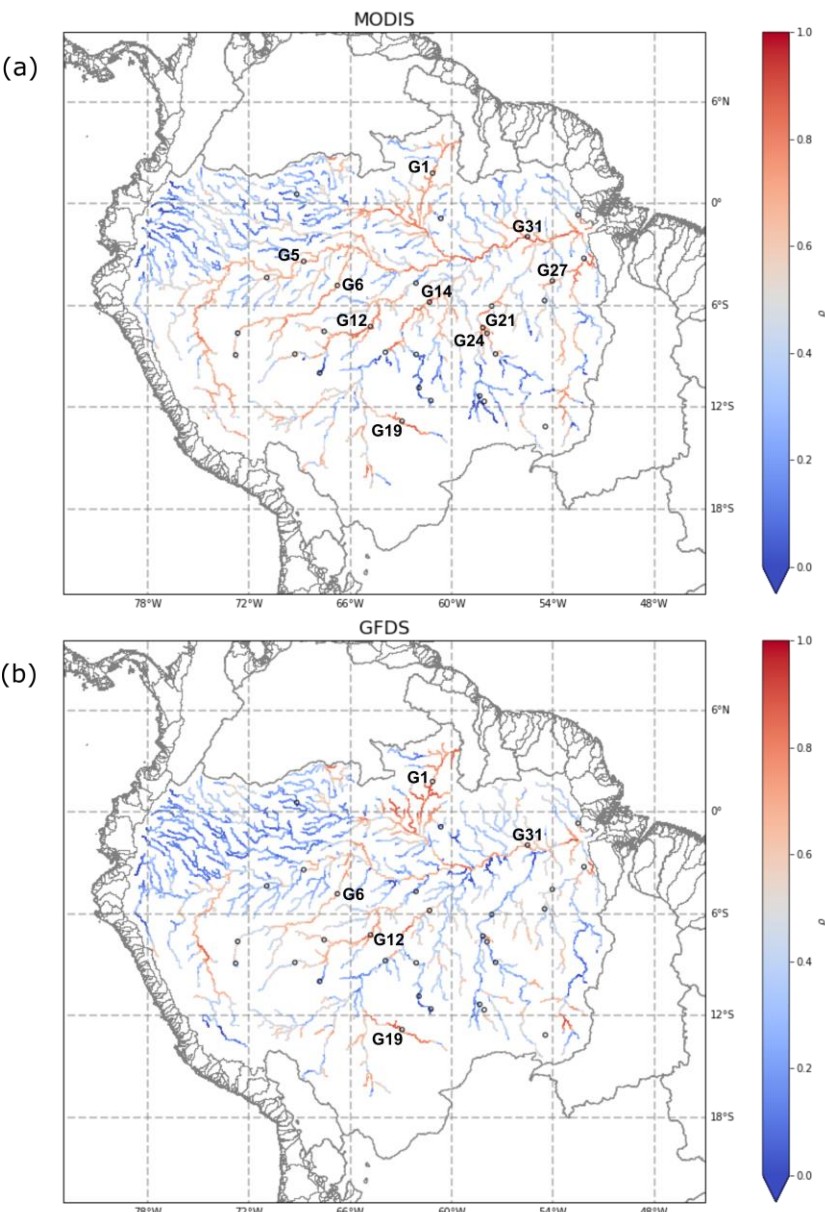

**Figure 5** Spearman correlation (ρ) between modelled river channel storage and MODIS (a) and GFDS (b) water extent using the *Optimal Grid Cell Selection* method (Method A) with a search window of 0.55°×0.55° (circle: gauging station; circle with label: potential SGRs sites where gauging data is available ).

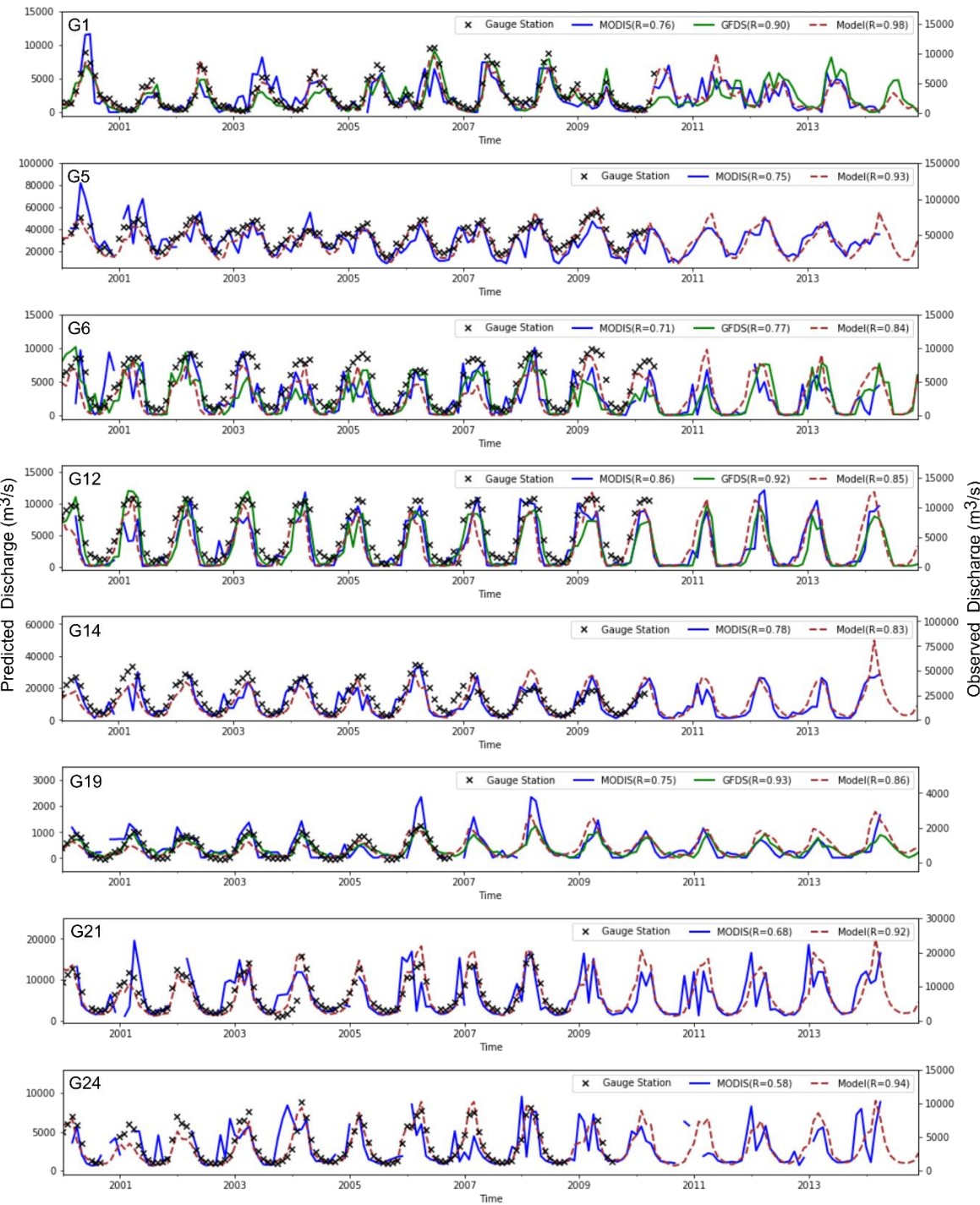

**Figure 6** Comparisons between observations (right axis) from gauging stations (black x) and river discharge estimates (left axis) derived using MODIS SGRs (blue line), GFDS SGRs (green line) and the W3 model (brown dash).

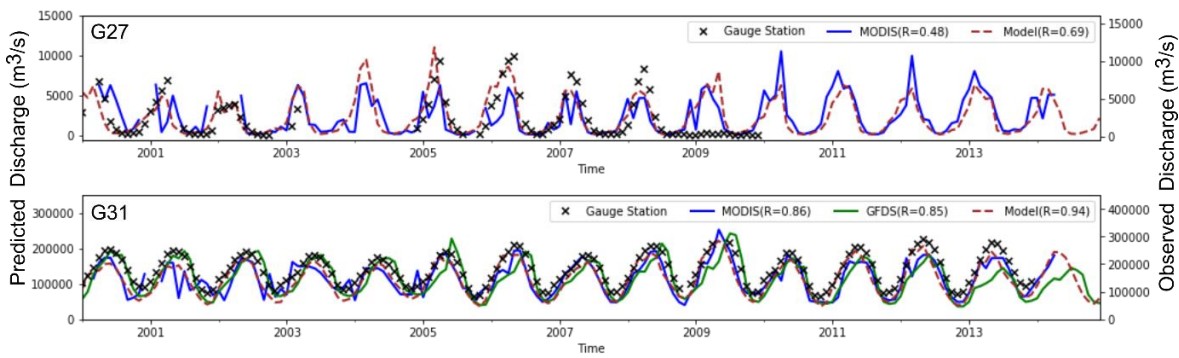

**Figure 6** Comparisons between observations (right axis) from gauging stations (black x) and river discharge estimates (left axis) derived using MODIS SGRs (blue line), GFDS SGRs (green line) and the W3 model (brown dash) (continued)

## 4 Discussion

The relationship between remote sensing signal, water extent, river channel storage and discharge enabled the estimation of river discharge from optical or passive microwave remote sensing. We showed that satellite gauging reaches (SGRs) can be developed without gauging station records, based on MODIS or GFDS water extents and W3 model estimated discharges. The *Optimal Selection* method (method A) with a search window of 0.55°×0.55° produced the best results. In total, we calculated Spearman correlations between modelled river channel storage and MODIS and GFDS water extent for 11752

grid cells across the Amazon Basin (Figure 3-5). The results suggest there are 3427 potential grid cells (ca. 17,135 km river reaches) to construct MODIS SGRs, and 1447 grid cells (ca. 7,235 km river reaches) to develop GFDS SGRs. The original MODIS data used in this research has a spatial resolution of 0.05° × 0.05°, which is higher than the GFDS data (0.09° × 0.1°). As such, MODIS should have better detection ability for river reaches with relatively small surface water extent. The performance of the method appears to be particularly related to the size of river reach. From upstream to downstream in the

Amazon Basin, $\rho$ between water extent and storage increases as river width increases, because MODIS and GFDS remote sensing are more sensitive to river reaches with larger surface water extent. Thus, the best locations for developing SGRs at the coarse resolution considered here are the lower reaches of the Amazon system.

The performance of SGRs over the Amazon Basin is generally good, as most river reaches have unregulated flows, and these

river reaches normally have wider river channels and large floodplains, as also remarked upon by Revilla-Romero et al. (2014). However, the performance of SGRs varies even for rivers of similar size. The relationship between water extent and storage or discharge also depends on local river characteristics and floodplain channel geometry (Moffitt et al., 2011; Brakenridge et al., 2012; Khan et al., 2012). Even though GFDS is suited for fewer river reaches than MODIS, the results showed that GFDS yielded better estimates of river discharge. A likely reason for this is that MODIS optical remote sensing

is limited to clear-sky conditions, whereas GFDS passive microwave remote sensing is much less affected by this. River

floodplains in the Amazon Basin are often covered with dense vegetation, and flood waters may spread below vegetation. Such flooding may be difficult to detect with optical imagery, but is still readily discernible with passive microwave remote sensing (Van Dijk et al., 2016). This is consistent with the results presented in Figures 3 and 4. In GFDS cases, the *Window Mean* method (method B) produced similar results to the *Optimal Selection* method (method A), but worse results in MODIS cases. We suspect that this is because some MODIS grid cells within the search window are influenced by clouds or forest.

Gauging stations are usually located in single, narrow and stable river reaches, while SGRs can be constructed in multiple, broad, and unstable river reaches provided variations can be detected by remote sensing. With that caveat, there were less than 1/3 of gauged river reaches that were feasible to develop MODIS SGRs and 1/6 to construct GFDS SGRs. Limited validation reaches with gauging stations does imply an underestimate of the percentage of successful SGRs. We focused on qualitative analysis rather than quantitative analysis for the performance of SGRs and the model. Qualitative analysis, such as Pearson correlation and Spearman's rank correlation, indicates the degree to which the estimations and observations show the same relative patterns while quantitative analysis, such as RMSE, reflects the differences between estimations and observations. Here the SGRs are mainly based on the model, so we would expect the developed SGRs should have the ability to reflect flow patterns rather than absolute flow values because of model biases. Tolerable errors and bias are contingent on the application for the data. For instance, for near real-time drought/flood monitoring it may be sufficient to know relative flows, whereas for water resources assessments, users require estimates that are bias-free estimates as much as possible. For the 10 gauging stations analysed here, the model showed a bias between -53% and 57% compared to the gauge records, with a median of -35%. This model bias propagates into the SGR estimates but could be removed easily where in situ data are available.

Based on comparison between gauging station records and river discharge estimates from MODIS, GFDS and the W3 model for period of 2000-2014, we conclude that when the W3 model performs quite well in terms of river discharge estimation, then SGRs can perform with a similar level of accuracy. In certain cases, the SGRs were able to perform better than the W3 model in reproducing the timing of peak flows. For instance, at gauging station G19, the satellite-derived peak flows from both MODIS and GFDS over the period 2000-2005 were closer to gauged peak river discharges than those estimated by the W3 model (Figure 6). However, there are also instances where the SGR estimates of discharge are inferior to those produced by the W3 model, e.g., for gauging stations G21 and G24. It is possible that in these instances MODIS has failed to measure water extent in small rivers or was affected by cloud cover. In other cases we suspect that poor results are attributable to data errors. For instance, the discharge observations at gauging station G27 were extremely low from late 2008 to 2009, suggesting a gauge measurement error or other artefact. Other performance problems may be attributable to the calibration processes and period, which were necessarily short. If SGRs were calibrated during a dry period, they may fail to estimate river discharge well during a wet period (and vice versa). For example, at gauging station G27, the SGR was not able to estimate peak flows accurately for the wet years from 2005-2009, and then estimated much larger river discharges than the

model during the dry years 2010-2014. This would be avoided if the full period had been used for SGR construction, which would be a pragmatic approach for operational implementation but would prevent independent evaluation in the context of the present study.

Previous research demonstrated that both gauging data and hydrology modelling can be used to calibrate the remote sensing signal for estimating river discharge (Brakenridge et al., 2012; Revilla-Romero et al., 2014). Van Dijk et al. (2016) developed gauge-based SGRs using optical and passive microwave derived water extent observations, which is valuable to gap-fill and extend gauging discharge records. In addition to that, we demonstrated that SGRs can be also developed using hydrological modelling. We compared our model-based SGRs to gauge-based SGRs from previous research (Van Dijk et al.,

2016) for all gauging reaches except gauging station G27 due to its unreliable records (Table 3). Both gauge-based and model-based GFDS SGRs at gauging station G12 and G19 have higher Pearson correlations than the model, which suggests opportunities for data assimilation to improve the model. At gauging station G1, G5, G21, and G24, the model performs much better than both gauge-based and model-based SGRs, which suggests that uncertainties in SGRs at these locations mainly arise from remote sensing, e.g. due to cloud and vegetation obstruction. Errors and uncertainties of the model, such as

from input data, routing, and conceptual structure, can also affect the performance of SGRs. For instance, for GFDS SGRs at gauging station G6 and MODIS SGRs at gauging station G31, gauge-based SGRs produced higher Pearson correlations than model-based SGRs. Compared to gauge-based SGRs, the main advantage of our method is the practical applicability in both gauged and ungauged rivers. Our results show that the model outperforms SGRs in most cases. Nonetheless, we consider SGRs as an alternative, simple and automated approach for river discharge prediction using satellite observation only. SGRs

would be useful as an alternative if the model was unable to provide real-time estimates, e.g. due to delayed rainfall gauge observations. As we used a model to train SGRs, poor model simulations might reduce the performance of SGRs. If more accurate and reliable hydrological models are available, SGRs can be redeveloped to estimate river discharge with greater accuracy. Overall, SGRs performed well in the current case study of the Amazon Basin. The W3 model, MODIS and GFDS remote sensing all provide information with global coverage. Therefore, there is further potential to develop satellite-based

river gauging elsewhere.

**Table 3** Performance comparisons between gauge-based SGRs, model-based SGRs and the W3 model (Pearson correlations between predicted and observed discharges).

| | | G1 | G5 | G6 | G12 | G14 | G19 | G21 | G24 | G31 | Mean |
|---|---|---|---|---|---|---|---|---|---|---|---|
| **MODIS SGRs** | **Gauge-based** | 0.75 | 0.77 | 0.74 | 0.86 | 0.77 | 0.88 | 0.48 | 0.6 | 0.92 | 0.75 |
| | **Model-based** | 0.76 | 0.75 | 0.71 | 0.86 | 0.78 | 0.75 | 0.68 | 0.58 | 0.86 | 0.75 |
| **GFDS SGRs** | **Gauge-based** | 0.88 | | 0.85 | 0.96 | | 0.95 | | | 0.85 | 0.9 |
| | **Model-based** | 0.9 | | 0.77 | 0.92 | | 0.93 | | | 0.85 | 0.87 |
| **Model** | | 0.98 | 0.93 | 0.84 | 0.85 | 0.83 | 0.86 | 0.92 | 0.94 | 0.94 | 0.9 |

The further development of the SGR methodologies could benefit from combining optical and passive microwave remote sensing. With higher spatial resolution, optical remote sensing is more suitable for measuring surface water extent in reaches without dense vegetation and when clear-sky conditions prevail. Passive microwave remote sensing, however, compensates for the limitations of optical remote sensing, but suffers from having lower spatial resolution. The main constraint in developing SGRs in this study was that the spatial resolutions of both MODIS and GFDS data were not high enough to detect changes in river dynamics in small rivers. New satellite imagery emerging from Sentinel-1 and Sentinel-2 provides further opportunities to develop satellite-based river gauging at a global scale. The spatial resolution of Sentinel-1 reaches to 5 m with C-band Synthetic Aperture Radar (C-SAR) working in all weather and in both day and night time conditions. The Sentinel-2 A and B multispectral instruments (MSI) have 13 spectral bands at 10-60 m spatial resolution and combined with Landsat observations means that revisit times in the order of days are now achievable. These developments offer great promise for the future development of SGRs and will be the subject of our on-going research.

## 5 Conclusions

We proposed and tested two methods for relating MODIS and GFDS-derived water extent to modelled river channel storage. For the Amazon Basin, river reaches with Spearman's rank correlation ($\rho$) between water extent and storage exceeding 0.6 were identified as suitable sites for developing SGRs. SGRs were then constructed across the Amazon Basin based on MODIS and GFDS water extent and modelled discharge, and river discharge estimates were evaluated using in situ river discharge measurements at 10 stations. Our main conclusions are:

(1) The *Optimal Grid Cell Selection* method performed better than the *Window Mean* method which related W3 model simulated river storage and discharge to MODIS and GFDS-derived surface water extent fraction, and a window size of 0.55°×0.55° was considered as a reasonable window for identifying the best remote sensing pixels for each model grid cell.

(2) There were strong correlations between modelled storage and both MODIS and GFDS water extent across the Amazon Basin. The *Optimal Selection* method is mainly limited by the size of river reach, as correlation generally increased from upstream to downstream as river width increased.

(3) In total, 17,135 km of river reaches in the Amazon Basin were assessed as suitable for constructing MODIS SGRs, and 7,235 km of river reaches were deemed suitable for developing GFDS SGRs. The best locations for developing SGRs were mostly situated in the lower channels of the Amazon River and its main tributaries.

(4) There were more potential SGRs derived using MODIS than GFDS, most likely because MODIS has higher spatial resolution than GFDS. However, GFDS SGRs predicted river discharges with more accuracy as GFDS was much less affected by cloud and dense vegetation than MODIS.

(5) Although the W3 model performed very well in terms of river discharge estimates in the Amazon Basin, MODIS and GFDS SGRs can still be useful for estimating river discharge in the absence of a real time hydrological model or gauging stations.

(6) SGRs are suitable for automated development at a global scale. Remote sensing with higher spatial resolution can help improve river discharge estimation capabilities of SGRs. This also creates potential opportunities to assimilate remote sensing observations, or derived discharge estimates, into hydrological model to improve river discharge estimation, and based on these, streamflow forecasts.

**Acknowledgements:**

The authors acknowledge the Global Runoff Data Centre (GRDC) and the U.S. Geological Survey (USGS) for providing the in situ river discharge measurement, and Hylke Beck and Aiguo Dai for compiling the discharge data. The first author thanks the ANU-CSC (the Australian National University and the China Scholarship Council) Scholarship for supporting his PhD study at the Australian National University. Calculations were performed on the high-performance computing system, Raijin, from the National Computational Infrastructure (NCI), supported by the Australian Government's National Collaborative Research Infrastructure Strategy (NCRIS). We also thank Florian Pappenberger, Christel Prudhomme, and three anonymous reviewers for their helpful suggestions that improved the manuscript.

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
