# Peer review of "Using modelled discharge to develop satellite-based river gauging: a case study for the Amazon Basin"

_Hydrology and Earth System Sciences, 2018_

## Referee Comment (RC1) · Anonymous Referee #1 · 28 Jun 2018

The study combined hydrological model outputs and satellite-derived surface inundation to develop satellite-based river gauging. The idea is interesting and potentially very useful, but the approach does not seem to me an optimal one with severe restrictions.

(1) Page 3 Section 2: "The fundamental assumption in our methodology is that there exist strong, monotonic relationships between remote sensing signal, surface water extent, river channel storage, and river discharge." This assumption is the basis of the whole study while it may not be true for many cases due to low-quality inundation observations caused by cloud (for optical sensors) or dense vegetation (for both optical and microwave), reservoir regulations, hilly terrain and inhomogeneity of the study region. The approach developed under this assumption will inevitably find difficulties for global applications.

(2) Section 2.3.2 "A Spearman correlation > 0.6 in a grid cell (0.05 $^\circ$×0.05 $^\circ$) was used to identify a potential river reach for developing SGR". It seems a paradox to me. If the high correlation implies the good quality of model and satellite results, why not use model simulations alone? In other words, under such high correlations, satellite retrievals do not provide much support on improving the model predictions.

(3) Why not use assimilation techniques and refine the model predictions by incorporating the satellite-derived inundation data?

(4) Overall the gauge readings are well correlated with satellite inundation data for the Amazon region (Pham-Duc et al., 2017). It will be interesting to check one more major river basin for evaluating the method.

(5) There are several satellite inundation data sets (e.g. long-term record described in Pham-Duc et al., 2017). What about the alternative choices of using these data sets?

Minor issue: Page 12 "A likely reason for this is that MODIS optical remote sensing is limited to clear-sky conditions and requires surface water to be unobscured by a dense vegetation canopy while GFDS passive microwave remote sensing is not affected by either of these factors". This statement is not accurate since passive microwave also has its limitations when sensing land surface over severer weather or dense vegetation conditions.

---

## Referee Comment (RC2) · C. Prudhomme (Referee) · 2 Jul 2018

General

The paper describes a method to use Earth Observations as way of estimating river discharge in ungagged basin. The method is applied to the Amazon river basin, and performance evaluated with a set of independent river gauged records.

The paper is very well written and clear, and aims to propose a solution to the challenge of lack of hydrological measurements in many parts of the world. The solution makes use of satellite imagery, which is great as more data are becoming available from the

[Figure]

EO community for hydrological applications. My recommendation is for publication after clarifications on a few points.

Major comments

Overall, the methods description is clear, and results show promising potential. However, I would have liked to see a commentary on the limitations of the method, in particular related to the use of hydrological model simulations to train the derivation of 'Satellite Gauging Reaches' SRGs. This is particular important in regions with no river gauged observations, as lack of observations can dramatically reduce hydrological modelling performance as no calibration can be undertaken. I believe the authors should also clarify their comment line 334-335 as this seams a circular argument, with hydrological simulations used to derive SGRs, and then SGRs used to calibrate hydrological models.

Another point of clarification regards the justification of the optimisation/ training strategy: SGRs discharge estimates are based on a built relationship between water extent and modelled channel storage, and then another transformation from water extent to river discharge. I am not sure I follow why they are two independent methods, and why the water extent cannot be directly trained using simulated river discharge.

Whilst promising, the authors only found that the methods could be applied and evaluated over less than 1/3 of gauged rivers (and 1/6 for the GFDS method). It would be insightful to have a commentary on the overall applicability of the method, and ways of improvement.

Minor comments

- The Data and methods section does not contain information on the data used for the hydrological mode, in particular the source of rainfall and potential evaporation time series. A commentary on the calibration of the model, especially if it used any of the 31 river gauges considered in the study, would be an important addition.

- The description of the GFDS dataset is not very clear, in particular regarding the time step of the time series (4days or 1 day?).

- line 143: change 'resampled to 8-day averages' to 'averaged to 8 days'.

- Section 3.1: it would be useful to know on how many points the evaluation is conducted over (for fig 2 and 3).

- Lines 222-225. Can you please provide a more quantified metrics, for example the number of false attributions in relation with the window size?

- lines 231-232: The last sentence is presumably referring to GFDS: please clarify.

- Section 3.2: can you justify the use of a correlation threshold of 0.6? Please also remind the reader that the vertical axes in fig 5 are not the same for observations and simulations. It would also be important to comment on the relatively low number of sites where the method is judged 'applicable (about 1/3 for MODIS, and only 1/6 for GFDS). As GFDS shows a relatively better performance in reproducing river discharge time series than channel storage (fig 5), it might be useful to consider a slightly lower threshold for the overall performance analysis.

---

## Referee Comment (RC3) · Anonymous Referee #3 · 2 Jul 2018

This study focusses on the estimation of river discharge from remote sensing observations. The authors developed a methodology, called satellite gauging reaches (SGRs), to derive river discharge over the Amazon Basin from optical (MODIS) or passive microwave (GFDS) observations. Other attempts to retrieve river discharge from remote sensing can be found in the literature, as noted by the authors in the introduction. I would have appreciated some kind of comparison with existing methods (at least in terms of potential errors). Here, either MODIS or GFDS is used to calibrate a relationship between surface water extent and discharge modelled by the W3 hydrological model. Performances of two methods (Optimal Grid Cell Selection and Window Mean) and five window sizes are compared for both MODIS and GFDS. The method and

window size that provide the best results are then selected to apply the SGRs over a few locations (15) where correlation is high enough. Discharge derived from SGR and modelled discharge are finally compared to observations from in situ gauge stations. Results show that SGR provides reasonable discharge estimates.

The idea of SGR is quite interesting, but its performances should be evaluated more comprehensively, especially for potential application in ungauged regions (as suggested by the authors). For instance, how would SGR behave over pixels where the correlation between surface water extent and modelled discharge is low (pixels excluded from the study)? How does the method compare with existing ones? What are the main limitations?

The paper is quite well organized, but could be improved by: - explaining the concept of SGR in the introduction - providing a workflow scheme of the overall methodology - better justifying the interest and added value of SGR compared to existing methods and models

Minor comments:

P6L3-4. Is there any quantitative criterion which motivated this choice?

P6L12-13. Not clear

P7L30-31. Is it a quantitative or qualitative result?

Figure 4. Colors of SGR sites (purple and black) are not clearly visible.

Figure 5. The figure is too small.

P12L14. Not clear

P12L29-30. Is it because of model biases?

---

## Referee Comment (RC4) · Anonymous Referee #4 · 4 Jul 2018

In my opinion, this research topic is very important, and the authors have made great efforts in impoving the technique of discharge estimation by combining the outputs from model and satellite products. However, the presentation and organization of the paper is very weak (although good grammar), make the readers very difficult to catch the key points and major conclusions. When you write a paper, you should consider the major conclusion/outputs, and the materials/work that can support the conclusion, making your "take-home-message" very clear;

[Figure]

while not presenting all the subtle details in the manuscript. In addition, this

paper only gives 5 figures, I would like to suggest adding quite a few tables to

better summarize the contents (including your method, datasets, evaluation

procedure, and so on).

Major comments: 1) suggest rewriting the abstract, to make it more clear by only summarizing the

major results and key conclusions. Your methodology should be described with

better clarity. 2) suggest removing much details in experiment setting, or use tables to list

different methods/scenarios, and your calibration/validation periods.

---

## Author Comment (AC1) · 21 Aug 2018

We thank the reviewer for the time and effort spent on the review of our manuscript. The reviewer provided us with helpful comments, which will greatly improve our manuscript. Below please find our response to reviewer's comments in detail.

Comment #1

"Page 3 Section 2: "The fundamental assumption in our methodology is that there exist strong, monotonic relationships between remote sensing signal, surface water extent, river channel storage, and river discharge." This assumption is the basis of the whole

study while it may not be true for many cases due to low-quality inundation observations caused by cloud (for optical sensors) or dense vegetation (for both optical and microwave), reservoir regulations, hilly terrain and inhomogeneity of the study region. The approach developed under this assumption will inevitably find difficulties for global applications."

Response #1

Our assumption really combines two hypotheses. One is that surface water extent is monotonically related to river channel storage and discharge. Another is that the remote sensing signal monotonically relates to surface water extent. It is true that the second hypothesis can be affected by cloud and dense vegetation, which was discussed in P12L16-24. It is also correct that reservoirs can affect the first hypothesis, but we are not sure that influence of hilly terrain and inhomogeneity of the study region on either hypothesis is to be expected.

Comment #2

"Section 2.3.2 "A Spearman correlation > 0.6 in a grid cell (0.05 °×0.05 °) was used to identify a potential river reach for developing SGR". It seems a paradox to me. If the high correlation implies the good quality of model and satellite results, why not use model simulations alone? In other words, under such high correlations, satellite retrievals do not provide much support on improving the model predictions."

Response #2

We appreciate the reviewer's comment. We do not believe this is a paradox as the high correlation here only implies the relationship between water extent and discharge is robust enough to develop SGRs in a certain river reach. In addition, our results in Section 3.2 indeed show the model outperforms SGRs in most cases. Thus, instead of using model, we consider SGRs as an alternative, simple and automated approach for river discharge prediction using satellite observation only. For example, SGR would be

useful as an alternative if the model was unable to provide real-time estimations due to delayed rainfall estimates. We will add these points to the discussion.

Comment #3

"Why not use assimilation techniques and refine the model predictions by incorporating the satellite-derived inundation data?"

Response #3

We agree that data assimilation is a promising way to refine model prediction, though it is not a solution without challenges of its own. It, too, requires a reasonable model in the first place, and then requires that satellite observations are even better than model simulations. Our study helps assess where this is the case. We intend to look at data assimilation in the future, but it is beyond the scope of this study.

Comment #4

"Overall the gauge readings are well correlated with satellite inundation data for the Amazon region (Pham-Duc et al., 2017). It will be interesting to check one more major river basin for evaluating the method."

Response #4

In response to the reviewer, we have evaluated SGRs elsewhere. For example, a SGR in the lower channel of the Niger Basin is shown in Figure A below. This is one example for another major river basin, but we do not propose to include it in the paper as it could be construed as selective. However, we do hope to undertake a global scale analysis of SGRs along the ones of this manuscript in future research.

Comment #5

"There are several satellite inundation data sets (e.g. long-term record described in Pham-Duc et al., 2017). What about the alternative choices of using these data sets?"

Response #5

The reason we chose MODIS and GFDS water extent data sets is primarily that they were used in a previous paper with good success. We thank the reviewer for suggesting these alternative data sets and had a look at them. However, as we found they are much coarser, we suspect they will not be more useful.

Comment #6

"Page 12 "A likely reason for this is that MODIS optical remote sensing is limited to clear-sky conditions and requires surface water to be unobscured by a dense vegetation canopy while GFDS passive microwave remote sensing is not affected by either of these factors". This statement is not accurate since passive microwave also has its limitations when sensing land surface over severer weather or dense vegetation conditions."

Response #6

Agreed. We will change P12L17-19 from "is not affected" to "is much less affected".

[Figure]

[Figure]

**Figure A** Comparison between observations (right axis) from gauging station (black x) and river discharge estimates (left axis) derived using MODIS SGR (blue line), GFDS SGR (green line) and the W3 model (brown dash) in the lower channel of the Niger Basin.

---

## Author Comment (AC2) · 21 Aug 2018

We thank the reviewer for her detailed comments and suggestions, which will greatly help us to improve our manuscript. Below please find our response to reviewer's comments in detail.

Comment #1

"Overall, the methods description is clear, and results show promising potential. However, I would have liked to see a commentary on the limitations of the method, in particular related to the use of hydrological model simulations to train the derivation

of 'Satellite Gauging Reaches' SRGs. This is particular important in regions with no river gauged observations, as lack of observations can dramatically reduce hydrological modelling performance as no calibration can be undertaken."

Response #1

We certainly agree with the reviewer that a poor model imposes its limitations on SGRs as we used a model to train SGRs. We will add further discussion of the limitations of SGRs caused by the model in the revised manuscript.

Comment #2

"I believe the authors should also clarify their comment line 334-335 as this seams a circular argument, with hydrological simulations used to derive SGRs, and then SGRs used to calibrate hydrological models."

Response #2

We agree that the reviewer is essentially correct that this is a circular argument. We will delete this sentence.

Comment #3

"Another point of clarification regards the justification of the optimisation/ training strategy: SGRs discharge estimates are based on a built relationship between water extent and modelled channel storage, and then another transformation from water extent to river discharge. I am not sure I follow why they are two independent methods, and why the water extent cannot be directly trained using simulated river discharge."

Response #3

Thank you for your comment. As there is a linear and direct relationship between river channel storage and discharge within the W3 model structure, water extent can be related to either river channel storage or discharge (see P5L8-9) with identical results. We chose storage for conceptual reasons.

Comment #4

"Whilst promising, the authors only found that the methods could be applied and evaluated over less than 1/3 of gauged rivers (and 1/6 for the GFDS method). It would be insightful to have a commentary on the overall applicability of the method, and ways of improvement."

Response #4

Agreed. The overall applicability of the method was described in P12L5-10, but we will add more discussion on the importance of river size, which appears the main determinant.

Some ways for improvement were described in the last paragraph of discussion section, which emphasized the use of much higher resolution satellite imagery, e.g. from Sentinel-1 and Sentinel-2.

Comment #5

"The Data and methods section does not contain information on the data used for the hydrological mode, in particular the source of rainfall and potential evaporation time series. A commentary on the calibration of the model, especially if it used any of the 31 river gauges considered in the study, would be an important addition."

Response #5

The model was not calibrated against the 31 river gauges. We will add further details on model inputs and calibration in the revised manuscript.

Comment #6

"The description of the GFDS dataset is not very clear, in particular regarding the time step of the time series (4days or 1 day?)."

Response #6

We apologise for the confusion, and will change these sentences to clarify the description of the GFDS dataset.

Comment #7

"line 143: change 'resampled to 8-day averages' to 'averaged to 8 days'."

Response #7

Agreed. We will change this sentence as suggested.

Comment #8

"Section 3.1: it would be useful to know on how many points the evaluation is conducted over (for fig 2 and 3)."

Response #8

Thank you for your suggestion. We calculated Spearman correlations between modelled river channel storage and MODIS and GFDS water extent for 11752 grid cells across the Amazon Basin. We will add a sentence about the number of points at which the evaluation was conducted.

Comment #9

"Lines 222-225. Can you please provide a more quantified metrics, for example the number of false attributions in relation with the window size?"

Response #9

Thank you for your suggestion. We will provide quantified metrics related to comment #8. Our results suggest there are 3427 potential grid cells (ca. 17,135 km river reaches) to construct MODIS SGRs, and 1447 grid cells (ca. 7,235 km river reaches) to develop GFDS SGRs.

Comment #10

"lines 231-232: The last sentence is presumably referring to GFDS: please clarify."

Response #10

Agreed. We will rephrase this.

Comment #11

"Section 3.2: can you justify the use of a correlation threshold of 0.6? Please also remind the reader that the vertical axes in fig 5 are not the same for observations and simulations. It would also be important to comment on the relatively low number of sites where the method is judged 'applicable (about 1/3 for MODIS, and only 1/6 for GFDS). As GFDS shows a relatively better performance in reproducing river discharge time series than channel storage (fig 5), it might be useful to consider a slightly lower threshold for the overall performance analysis."

Response #11

Thank you for these suggestions. In the caption of Figure 5, we mentioned that observations from gauging stations are shown on the right axis and river discharge estimates derived using remote sensing and model on the left axis. We will try to explain this better.

Gauging stations are usually located in single, narrow and stable river reaches, while SGRs can be constructed in multiple, broad, and unstable river reaches provided variations can be detected by remote sensing. With that caveat, there were less than 1/3 of gauged river reaches that were feasible to develop MODIS SGRs and 1/6 to construct GFDS SGRs. Limited validation reaches with gauging stations does imply an underestimate of the percentage of successful SGRs. We will add sentences to explain the relatively low number of sites where the method was judged applicable.

The correlation threshold of 0.6 is an empirical threshold we used to distinguish potential river reaches to develop SGRs. If we develop SGRs where correlations are below 0.6, the overall performance would worsen accordingly. For example, the correlation between modelled storage and remote sensing water extent at gauging station G10 was 0.48 for MODIS and 0.58 for GFDS. When we constructed SGRs, the performance of MODIS and GFDS SGRs reached low values of 0.33 and 0.36, respectively. Thus, we propose to keep 0.6 as the threshold.

―――――――――――――――――――

---

## Author Comment (AC3) · 21 Aug 2018

We thank the reviewer for the thoughtful comments and constructive suggestions, which will help us to improve the quality of the manuscript. Below please find our response to reviewer's comments in detail.

Comment #1

"The idea of SGR is quite interesting, but its performances should be evaluated more comprehensively, especially for potential application in ungauged regions (as suggested by the authors). For instance, how would SGR behave over pixels where the

correlation between surface water extent and modelled discharge is low (pixels excluded from the study)? How does the method compare with existing ones? What are the main limitations?"

Response #1

We thank the reviewer for these valuable suggestions. We would like to make it clear that the gauging data are only used for validation and evaluation. Therefore, even river reaches with gauging stations can effectively be considered as examples of ungauged regions. In truly ungauged regions, we would not be able to evaluate the SGR method.

We did not exclude any pixels in this study. In the Window Mean method, we used all the pixels within a window, but we did not select all the pixels as target pixels because some of them did not have enough modelled discharge. In the Optimal Selection method, we picked the best pixel in a window, we did not use others as they were not the most suitable ones to develop SGRs.

We compared our model-based SGRs to gauge-based SGRs from previous research (Van Dijk et al., 2016), which can help analyse the sources of uncertainties of model-based SGRs (Table A below). Both gauge-based and model-based GFDS SGRs at gauging station G12 and G19 have higher Pearson correlations than the model, which suggests value to implement data assimilation to improve the model. At gauging station G1, G5, G21, and G24, the model behaves much better than both gauge-based and model-based SGRs, which suggests that uncertainties in SGRs at these locations are mainly coming from remote sensing. Errors and uncertainties of the model, such as from input data, routing, and conceptual structure, can also affect the performance of SGRs. For instance, for GFDS SGRs at gauging station G6 and MODIS SGRs at gauging station G31, gauge-based SGRs produced higher Pearson correlations than model-based SGRs. We will add this comparison with existing method in the revised manuscript.

We will add more discussion about the main limitations of SGRs (see response to
comment #4 from reviewer #2).

Comment #2

"The paper is quite well organized, but could be improved by: - explaining the concept of SGR in the introduction - providing a workflow scheme of the overall methodology - better justifying the interest and added value of SGR compared to existing methods and models."

Response #2

Thank you for your suggestions. We will change the introduction to explain the concept of SGRs better.

We will also add a figure with a workflow of the overall methodology in Figure B below.

We will add discussion of our method compared to existing methods and models in the revised manuscript. Previous research demonstrated that both gauging data and hydrology modelling can be used to calibrate the remote sensing signal for estimating river discharge (Brakenridge et al., 2012; Revilla-Romero et al., 2014). Van Dijk et al. (2016) developed gauge-based SGRs using optical and passive microwave derived water extent observations, which is valuable to gap-fill and extend gauging discharge records. In addition to that, we prove that SGRs can be also developed using hydrological modelling. Compared to gauge-based SGRs, the main advantage of our method is the practical applicability in both gauged and ungauged rivers. Our results show the model outperforms SGRs in most cases. Nonetheless, we consider SGRs as an alternative, simple and automated approach for river discharge prediction using satellite observation only. For example, SGR would be useful as an alternative if the model was unable to provide real-time estimations due to delayed rainfall estimates. If more accurate and reliable hydrological models are available, SGRs can be redeveloped to estimate river discharge with greater accuracy.

Comment #3

"P6L3-4. Is there any quantitative criterion which motivated this choice?"

Response #3

$10^2$ m3s-1 was an empirical threshold that we chose as it shows the main river network in the Amazon Basin (Figure 1). We assume that rivers with mean discharge above $10^2$ m3s-1 have wider channels and broader floodplains than those with mean discharge below this threshold. In addition, annual mean discharge at all gauging stations ranges from 235 to 172,167 m3s-1 except G29 with very low discharge of 84 m3s-1. Thus, this threshold seemed a reasonable choice to cover river reaches with gauging stations for validation and evaluation.

Comment #4

"P6L12-13. Not clear"

Response #4

Agreed. We will change this sentence to make it clear.

Comment #5

"P7L30-31. Is it a quantitative or qualitative result?"

Response #5

The Optimal Selection method was chosen based on quantitative analysis as it shows highest correlations compared to the Window Mean method, while the selection of window size was decided based on qualitative interpretation.

Comment #6

"Figure 4. Colors of SGR sites (purple and black) are not clearly visible."

Response #6

Thank you for your suggestion. We will improve this figure (Figure 4 below).

Comment #7

"Figure 5. The figure is too small."

Response #7

Agreed. We will enlarge it to cover two pages.

Comment #8

"P12L14. Not clear"

Response #8

We will change this sentence to make it clearer.

Comment #9

"P12L29-30. Is it because of model biases?"

Response #9

Correct (see P12L32-P13L2). We will explain this.

Reference

Brakenridge, G. R., Cohen, S., Kettner, A. J., De Groeve, T., Nghiem, S. V., Syvit-ski, J. P. M., and Fekete, B. M.: Calibration of satellite measurements of river discharge using a global hydrology model, Journal of Hydrology, 475, 123-136, doi:10.1016/j.jhydrol.2012.09.035, 2012.

Revilla-Romero, B., Thielen, J., Salamon, P., De Groeve, T., and Brakenridge, G. R.: Evaluation of the satellite-based Global Flood Detection System for measuring river discharge: influence of local factors, Hydrology and Earth System Sciences, 18, 4467-4484, doi:10.5194/hess-18-4467-2014, 2014.

Van Dijk, A. I. J. M., Brakenridge, G. R., Kettner, A. J., Beck, H. E., De Groeve, T., and Schellekens, J.: River gauging at global scale using optical and

passive microwave remote sensing, Water Resources Research, 52, 6404-6418, doi:10.1002/2015WR018545, 2016.

[Figure]

**Table A** Performance comparison between gauge-based SGRs, model-based SGRs and the W3 model
(Pearson correlations between predicted and observed discharges).

| | | G1 | G5 | G6 | G12 | G14 | G19 | G21 | G24 | G31 | Mean |
|---|---|---|---|---|---|---|---|---|---|---|---|
| **MODIS SGRs** | **Gauge-based** | 0.75 | 0.77 | 0.74 | 0.86 | 0.77 | 0.88 | 0.48 | 0.60 | 0.92 | 0.75 |
| | **Model-based** | 0.76 | 0.75 | 0.71 | 0.86 | 0.78 | 0.75 | 0.68 | 0.58 | 0.86 | 0.75 |
| **GFDS SGRs** | **Gauge-based** | 0.88 | | 0.85 | 0.96 | | 0.95 | | | 0.85 | 0.90 |
| | **Model-based** | 0.90 | | 0.77 | 0.92 | | 0.93 | | | 0.85 | 0.87 |
| **Model** | | 0.98 | 0.93 | 0.84 | 0.85 | 0.83 | 0.86 | 0.92 | 0.94 | 0.94 | 0.90 |

[Figure]

**Figure B** Workflow of the overall methodology (rectangle: data; diamond: method; parallelogram: validation).

[Figure]

**Figure 4** Spearman correlation (ρ) between modelled river channel storage and MODIS (a) and GFDS
(b) water extent using the Optimal Grid Cell Selection method (Method A) with a search window of
0.55°×0.55° (circle: gauging station; circle with black label: potential SGRs sites where gauging data is
available).

[Figure]

**Figure 4** Spearman correlation (ρ) between modelled river channel storage and MODIS (a) and GFDS (b) water extent using the Optimal Grid Cell Selection method (Method A) with a search window of 0.55°×0.55° (circle: gauging station; circle with black label: potential SGRs sites where gauging data is available). (continued)

---

## Author Comment (AC4) · 21 Aug 2018

We thank the reviewer for taking the time and effort to review our manuscript thoroughly. The reviewer provided us with valuable comments, which will greatly improve our manuscript. Below please find our response to reviewer's comments in detail.

Comment #1

"suggest rewriting the abstract, to make it more clear by only summarizing the major results and key conclusions. Your methodology should be described with better clarity."

Response #1

We thank the reviewer for this constructive suggestion, and will rewrite the abstract as suggested.

We will also add a workflow to explain our methodology better (see response to comment #2 from reviewer #3).

Comment #2

"suggest removing much details in experiment setting, or use tables to list different methods/scenarios, and your calibration/validation periods."

Response #2

Thank you for these suggestions. We will use two tables to help describe our experiments (Table B and C).

**Table B** Experiment design (window size) for two methods to develop SGRs

| Experiments | I | II | III | IV | V |
|---|---|---|---|---|---|
| Optimal Selection | 0.15° | 0.35° | 0.55° | 0.75° | 0.95° |
| Window Mean | 0.15° | 0.35° | 0.55° | 0.75° | 0.95° |

**Table C** Training and validation periods for cross-validation method

| Periods | I | II | III |
|---|---|---|---|
| Training Period | 2005-2014 | 2000-2004&2010-2014 | 2000-2009 |
| Validation Period | 2000-2004 | 2005-2009 | 2010-2014 |

---

## Author Response (AR1)

**Response to Reviewers**

**''Using modelled discharge to develop satellite-based river gauging: a case study for the Amazon Basin'' by Jiawei Hou et al.**

We thank the editor and all the reviewers for their constructive comments, which enabled us to greatly improve the quality of our manuscript. The editor and reviewers' comments are reproduced below in blue, and our responses are presented in black.

**Editor Comments:**

(1) Your introduction talks about deriving ground data such as bathymetry and manning roughness. I believe in particular in the latter, you should acknowledge the uncertainties of this variable which can be done through an additional sentence and/or reference.

Agreed. We modified the sentence in P3L8-12 to

> "*However, in addition to remotely sensed data, additional field data including river depth and roughness coefficient are needed to apply this method and can introduce large uncertainties, which limits its predictive performance (Te Chow, 1959; LeFavour and Alsdorf, 2005; Jung et al., 2010; Woldemichael et al., 2010; Michailovsky et al., 2012).*"

(2) The current comparison does not take account of any uncertainties neither in the model nor in the satellite estimated discharges – I do believe that it would be valuable information to understand the individual uncertainties respectively.

To some extent our comparison does take account into uncertainties as we compared predicted discharges from SGRs and the model both to gauging data, which were independent. The errors and uncertainties of SGRs are partly coming from the model, such as input data, routing, and conceptual structure. For example, poor rainfall data will reduce the performance of the model, and the model may not represent some important features, such as large wetlands and reservoirs affecting the routing. Additionally, observing capabilities of remote sensing are limited by cloud and dense vegetation, which are also the sources of uncertainties of SGRs. We compared the performance differences between gauge-based SGRs and model-based SGRs to discuss where the uncertainties come from (also see response to comment #1 from reviewer #3). The following was added in P17L11-19,

> "*We compared our model-based SGRs to gauge-based SGRs from previous research (Van Dijk et al., 2016) for all gauging reaches except gauging station G27 due to its unreliable records (Table 3). Both gauge-based and model-based GFDS SGRs at gauging station G12 and G19 have higher Pearson correlations than the model, which suggests opportunities for data assimilation to improve the model. At gauging station G1, G5, G21, and G24, the model performs much better than both gauge-based and model-based SGRs, which suggests that uncertainties in*

I

*SGRs at these locations mainly arise from remote sensing, e.g. due to cloud or vegetation obstruction. Errors and uncertainties of the model, such as from input data, routing, and conceptual structure, can also affect the performance of SGRs. For instance, for GFDS SGRs at gauging station G6 and MODIS SGRs at gauging station G31, gauge-based SGRs produced higher Pearson correlations than model-based SGRs.*"

(3) I am surprised to read that you did not use nay daily data although they are available. I think you should at least analyse how your different products behave daily (and produce plot showing discharge time series for a view locations in the supplementary material).

Thank you for your suggestion. We have collected some additional daily gauging data, and put daily SGRs analysis in the Supplement. In the main text we added (P11L4-5):

"*The performance of daily, 8-day and monthly MODIS and GFDS SGRs are compared and discussed in the Supplement (Figure S1).*"

In the Supplementary Material we added the following:

"*We compared daily, 8-day and monthly performance of the MODIS and GFDS SGRs and the model in two river reaches in the lower channel of the Amazon River to evaluate the effect of temporal aggregation on prediction accuracy. The daily streamflow records were derived from Brazil's National Water Agency database by H.E. Beck (Princeton University, pers. comm.). To highlight temporal details, below we compared the different estimates for an arbitrary 13-month period (August, 2006 - August, 2007 for G32; June, 2006 - June, 2007 for G33). The daily and 8-day SGR predictions are noisy (Figure S1a, b, d and e) as the GFDS signal is influenced by many factors such as the weak magnitude of the radiance received at the passive sensor, the changing scanning geometry, footprint size of each swath, and the path of the radiation through the atmosphere (Kugler and De Groeve, 2007), whereas the MODIS signal sometimes appears to be affected by cloud or aerosols. However, both MODIS and GFDS SGRs can reflect monthly and seasonal discharge dynamics reliably (Figure S1c and f).*"

[Figure]

***Figure S1*** *Comparisons between observations (right axis) from gauging stations (black dash) and river discharge estimates (left axis) derived using MODIS SGRs (blue line), GFDS SGRs (green line) and the W3 model (brown line) (top row: daily results; middle row: 8-day results; bottom row: monthly results; left column: gauge station G32 (-3.06 ˚S, -59.65 ˚W); right column: gauge station G33 (-1.92 ˚S, -55.51 ˚W).*

**Reviewer #1 Comments:**

The study combined hydrological model outputs and satellite-derived surface inundation to develop satellite-based river gauging. The idea is interesting and potentially very useful, but the approach does not seem to me an optimal one with severe restrictions.

(1) Page 3 Section 2: "The fundamental assumption in our methodology is that there exist strong, monotonic relationships between remote sensing signal, surface water extent, river channel storage, and river discharge." This assumption is the basis of the whole study while it may not be true for many cases due to low-quality inundation observations caused by cloud (for optical sensors) or dense vegetation (for both optical and microwave), reservoir regulations, hilly terrain and inhomogeneity of the study region. The approach developed under this assumption will inevitably find difficulties for global applications.

Our assumption really combines two hypotheses. One is that surface water extent is monotonically related to river channel storage and discharge. Another is that the remote sensing signal monotonically relates to surface water extent. It is true that the second hypothesis can be affected by cloud and dense vegetation, which was and is discussed (P15L23-P16L7). It is also correct that reservoirs can affect the first hypothesis if they exist, but we are not sure that influence of hilly terrain and inhomogeneity of the study region on either hypothesis is to be expected.

(2) Section 2.3.2 "A Spearman correlation > 0.6 in a grid cell (0.05 °×0.05 °) was used to identify a potential river reach for developing SGR". It seems a paradox to me. If the high correlation implies the good quality of model and satellite results, why not use model simulations alone? In other words, under such high correlations, satellite retrievals do not provide much support on improving the model predictions.

We appreciate the reviewer's comment. We do not believe this is a paradox as the high correlation here only implies the relationship between water extent and discharge is robust enough to develop SGR in a certain river reach. In addition, our results in Section 3.2 indeed show the model outperforms SGRs in most cases. In order to explain the usefulness of SGRs compared to modelling, we added sentences in P17L20-25:

> "*Our results show that the model outperforms SGRs in most cases. Nonetheless, we consider SGR as an alternative, simple and automated approach for river discharge prediction using satellite observation only. For example, SGRs would be useful as an alternative if the model was unable to provide real-time estimations due to delayed rainfall estimates. As we used a model to train SGRs, poor model simulations might reduce the performance of SGRs. If more accurate and reliable hydrological models are available, SGRs can be redeveloped to estimate river discharge with greater accuracy.*"

(3) Why not use assimilation techniques and refine the model predictions by incorporating the satellite-derived inundation data?

We agree that data assimilation is a promising way to refine model prediction, though it is not without challenges of its own. It, too, requires a reasonable model in the first place, and then requires that satellite observations are even better than model simulations. Our study helps assess where this is the case, which was analysed based on comparison between gauge-based SGRs, model-based SGRs, and the model. We added a sentence about this analysis in P17L12-14:

> "*Both gauge-based and model-based GFDS SGRs at gauging station G12 and G19 have higher Pearson correlations than the model, which suggests opportunities for data assimilation to improve the model.*"

We intend to look at data assimilation in the future, but it is beyond the scope of this study.

(4) Overall the gauge readings are well correlated with satellite inundation data for the Amazon region (Pham-Duc et al., 2017). It will be interesting to check one more major river basin for evaluating the method.

In response to the reviewer, we evaluated SGRs elsewhere. For example, a SGR in the lower channel of the Niger Basin is shown in Figure A below. This is one example for another major river basin, but we do not propose to include it in the paper as it could be construed as selective or arbitrary. However, we do hope to undertake a global scale analysis of SGRs along the lines suggested in future.

[Figure]

**Figure A** Comparison between observations (right axis) from gauging station (black x) and river discharge estimates (left axis) derived using MODIS SGR (blue line), GFDS SGR (green line) and the W3 model (brown dash) in the lower channel of the Niger Basin.

(5) There are several satellite inundation data sets (e.g. long-term record described in Pham-Duc et al., 2017). What about the alternative choices of using these data sets?

The reason we chose MODIS and GFDS water extent data sets is primarily that they were used in a previous paper with good success. We thank the reviewer for suggesting these alternative data sets and carefully considered them. However, as we found they are much coarser, we concluded that they are not likely to produce better results.

(6) Page 12 "A likely reason for this is that MODIS optical remote sensing is limited to clear-sky conditions and requires surface water to be unobscured by a dense vegetation canopy while GFDS passive microwave remote sensing is not affected by either of these factors". This statement is not accurate since passive microwave also has its limitations when sensing land surface over severer weather or dense vegetation conditions.

Agreed. We modified this sentence in P15L24-P16L1 to:

> "*A likely reason for this is that MODIS optical remote sensing is limited to clear-sky conditions and requires surface water to be unobscured by a dense vegetation canopy, whereas GFDS passive microwave remote sensing is much less affected by either of these factors.*"

**Reviewer #2 Comments:**

The paper describes a method to use Earth Observations as way of estimating river discharge in ungagged basin. The method is applied to the Amazon River basin, and performance evaluated with a set of independent river gauged records.

The paper is very well written and clear, and aims to propose a solution to the challenge of lack of hydrological measurements in many parts of the world. The solution makes use of satellite imagery, which is great as more data are

becoming available from the EO community for hydrological applications. My recommendation is for publication after clarifications on a few points.

(1) Overall, the methods description is clear, and results show promising potential. However, I would have liked to see a commentary on the limitations of the method, in particular related to the use of hydrological model simulations to train the derivation of 'Satellite Gauging Reaches' SRGs. This is particular important in regions with no river gauged observations, as lack of observations can dramatically reduce hydrological modelling performance as no calibration can be undertaken.

We certainly agree with the reviewer that a poor model imposes its limitations on SGRs since we used a model to train SGRs. We acknowledged the limitations of the method and provided possible future ways to solve this issue by adding sentences in P17L23-25:

> "*As we used a model to train SGRs, poor model simulations might reduce the performance of SGRs. If more accurate and reliable hydrological models are available, SGRs can be redeveloped to estimate river discharge with greater accuracy.*"

(2) I believe the authors should also clarify their comment line 334-335 as this seams a circular argument, with hydrological simulations used to derive SGRs, and then SGRs used to calibrate hydrological models.

We agree that in a sense this is essentially a circular argument. We deleted this sentence in the revised manuscript.

(3) Another point of clarification regards the justification of the optimisation/ training strategy: SGRs discharge estimates are based on a built relationship between water extent and modelled channel storage, and then another transformation from water extent to river discharge. I am not sure I follow why they are two independent methods, and why the water extent cannot be directly trained using simulated river discharge.

Thank you for your comment. As there is a linear and direct relationship between river channel storage and discharge within the W3 model structure, water extent can be related to either river channel storage or discharge (see P6L3-6) with identical results. We chose storage for conceptual reasons.

(4) Whilst promising, the authors only found that the methods could be applied and evaluated over less than 1/3 of gauged rivers (and 1/6 for the GFDS method). It would be insightful to have a commentary on the overall applicability of the method, and ways of improvement.

Agreed. The overall applicability of the method was described in P16L5-17, but to highlight the main determinant of overall applicability, we added a sentence in P15L13-14:

> "*The performance of the method appears to be particularly related to the size of river reach.*"

and we also changed sentences in the conclusion section (P19L3-5 and P19L9-11) to:

*"There were strong correlations between modelled storage and MODIS and GFDS water extent across the Amazon Basin. The Optimal Selection method is mainly limited by the size of river reach, as correlation generally increased from upstream to downstream as river width increased."*

and

*"There were more potential SGRs derived using MODIS than GFDS, most likely because MODIS has higher spatial resolution than GFDS. However, GFDS SGRs predicted river discharges with more accuracy as GFDS was much less affected by cloud and dense vegetation than MODIS."*

Some ways for improvement were described in the last paragraph of discussion section, which emphasized the use of much higher resolution satellite imagery, e.g. from Sentinel-1 and Sentinel-2.

(5) The Data and methods section does not contain information on the data used for the hydrological mode, in particular the source of rainfall and potential evaporation time series. A commentary on the calibration of the model, especially if it used any of the 31 river gauges considered in the study, would be an important addition.

The model was not calibrated against the 31 river gauges. We added details on model calibration in P6L1-2:

*"This model was not calibrated against gauging data used in this study."*

(6) The description of the GFDS dataset is not very clear, in particular regarding the time step of the time series (4days or 1 day?).

We apologise for the confusion, and changed these sentences to clarify the description of the GFDS dataset in P5L7-12:

*"The GFDS raster data product used here, named 'merged 4-day average datasets', provides daily s as an average value of the signal for the current day and the signal from the last 3 days, with a spatial resolution of 0.09 °×0.1 ° over the period of 2000-2014."*

(7) line 143: change 'resampled to 8-day averages' to 'averaged to 8 days'.

Agreed. We changed this sentence in P6L2-3 to:

*"Daily simulated river channel storage and discharge in 0.05 °×0.05 ° grid cells were used in this research and averaged to 8 days to relate them to remote sensing data."*

(8) Section 3.1: it would be useful to know on how many points the evaluation is conducted over (for fig 2 and 3).

Thank you for your suggestion. We added a sentence about the number of points at which the evaluation was conducted in P15L8-10:

*"In total, we calculated Spearman correlations between modelled river channel storage and MODIS and GFDS water extent for 11752 grid cells across the Amazon Basin (Figure 3-5)."*

(9) Lines 222-225. Can you please provide a more quantified metrics, for example the number of false attributions in relation with the window size?

Thank you. We provided quantified metrics related to comment #8 in P15L10-11:

> "*The results suggest there are 3427 potential grid cells (ca. 17,135 km river reaches) to construct MODIS SGRs, and 1447 grid cells (ca. 7,235 km river reaches) to develop GFDS SGRs.*"

We also added a sentence in the conclusion section (P19L6-7):

> "*In total, 17,135 km of river reaches in the Amazon Basin were assessed as suitable for constructing MODIS SGRs, and 7,235 km of river reaches were deemed suitable for developing GFDS SGRs.*"

(10) lines 231-232: The last sentence is presumably referring to GFDS: please clarify.

Agreed. We rephrased this sentence in P9L7-9 to:

> "*For GFDS SGRs, there were more river reaches with low correlations ($\rho < 0.4$) in upstream tributaries, and the lower reach of the Amazon River did not show continuous high correlations ($\rho > 0.8$).*"

(11) Section 3.2: can you justify the use of a correlation threshold of 0.6? Please also remind the reader that the vertical axes in fig 5 are not the same for observations and simulations. It would also be important to comment on the relatively low number of sites where the method is judged 'applicable (about 1/3 for MODIS, and only 1/6 for GFDS). As GFDS shows a relatively better performance in reproducing river discharge time series than channel storage (fig 5), it might be useful to consider a slightly lower threshold for the overall performance analysis.

Thank you for these suggestions. In the caption of Figure 6, we mentioned that observations from gauging stations are shown on the right axis and river discharge estimates derived using remote sensing and model on the left axis. We added sentences to explain this better in P10L11-14:

> "*We focused on flow pattern comparisons between predicted and observed discharges, so different vertical axes were chosen to bring them close to each other (observations from gauging stations are shown on the right axis and river discharge estimates derived using remote sensing and model on the left axis).*"

We added sentences to explain the relatively low number of sites where the method was judged applicable in P16L9-12:

> "*Gauging stations are usually located in single, narrow and stable river reaches, while SGRs can be constructed in multiple, broad, and unstable river reaches provided variations can be detected by remote sensing. With that caveat, there were less than 1/3 of gauged river reaches that were feasible to develop MODIS SGRs and 1/6 to construct GFDS SGRs. Limited validation reaches with gauging stations does imply an underestimate of the percentage of successful SGRs.*"

The correlation threshold of 0.6 is an empirical threshold we used to distinguish potential river reaches to develop SGRs. If we would develop SGRs where correlations are lower or higher, the overall performance would worsen (improve) accordingly. For example, the correlation between modelled storage and remote sensing water extent at gauging station G10 was 0.48 for MODIS and 0.58 for GFDS. When we constructed SGRs, the performance of MODIS and GFDS SGRs produced values of 0.33 and 0.36, respectively. We propose to keep 0.6 as the threshold.

**Reviewer #3 Comments:**

This study focusses on the estimation of river discharge from remote sensing observations. The authors developed a methodology, called satellite gauging reaches (SGRs), to derive river discharge over the Amazon Basin from optical (MODIS) or passive microwave (GFDS) observations. Other attempts to retrieve river discharge from remote sensing can be found in the literature, as noted by the authors in the introduction. I would have appreciated some kind of comparison with existing methods (at least in terms of potential errors). Here, either MODIS or GFDS is used to calibrate a relationship between surface water extent and discharge modelled by the W3 hydrological model. Performances of two methods (Optimal Grid Cell Selection and Window Mean) and five window sizes are compared for both MODIS and GFDS. The method and window size that provide the best results are then selected to apply the SGRs over a few locations (15) where correlation is high enough. Discharge derived from SGR and modelled discharge are finally compared to observations from in situ gauge stations. Results show that SGR provides reasonable discharge estimates.

(1) The idea of SGR is quite interesting, but its performances should be evaluated more comprehensively, especially for potential application in ungauged regions (as suggested by the authors). For instance, how would SGR behave over pixels where the correlation between surface water extent and modelled discharge is low (pixels excluded from the study)? How does the method compare with existing ones? What are the main limitations?

We thank the reviewer for these valuable suggestions. We would like to make it clear that the gauging data are only used for validation and evaluation. Therefore, even river reaches with gauging stations can effectively be considered as examples of ungauged regions. In truly ungauged regions, we would not be able to evaluate the SGR method.

We did not exclude any pixels in this study. In the Window Mean method, we used all the pixels within a window, but we did not select all the pixels as target pixels because some of them did not have enough modelled discharge. In the Optimal Selection method, we picked the best pixel in a window, we did not use others as they were not the most suitable ones to develop SGRs.

We added analysis on comparison between our method and existing one in P17L11-19 and P18L4-6:

"*We compared our model-based SGRs to gauge-based SGRs from previous research (Van Dijk et al., 2016) for all gauging reaches except gauging station G27 due to its unreliable records (Table 3). Both gauge-based and model-*

IX

*based GFDS SGRs at gauging station G12 and G19 have higher Pearson correlations than the model, which suggests opportunities for data assimilation to improve the model. At gauging station G1, G5, G21, and G24, the model performs much better than both gauge-based and model-based SGRs, which suggests that uncertainties in SGRs at these locations mainly arise from remote sensing, e.g. due to cloud or dense vegetation. Errors and uncertainties of the model, such as from input data, routing, and conceptual structure, can also affect the performance of SGRs. For instance, for GFDS SGRs at gauging station G6 and MODIS SGRs at gauging station G31, gauge-based SGRs produced higher Pearson correlations than model-based SGRs.*"

**Table 3** *Performance comparisons between gauge-based SGRs, model-based SGRs and the W3 model (Pearson correlations between predicted and observed discharges).*

|  |  | *G1* | *G5* | *G6* | *G12* | *G14* | *G19* | *G21* | *G24* | *G31* | *Mean* |
|---|---|---|---|---|---|---|---|---|---|---|---|
| *MODIS SGRs* | *Gauge-based* | 0.75 | 0.77 | 0.74 | 0.86 | 0.77 | 0.88 | 0.48 | 0.6 | 0.92 | 0.75 |
|  | *Model-based* | 0.76 | 0.75 | 0.71 | 0.86 | 0.78 | 0.75 | 0.68 | 0.58 | 0.86 | 0.75 |
| *GFDS SGRs* | *Gauge-based* | 0.88 |  | 0.85 | 0.96 |  | 0.95 |  |  | 0.85 | 0.9 |
|  | *Model-based* | 0.9 |  | 0.77 | 0.92 |  | 0.93 |  |  | 0.85 | 0.87 |
| *Model* |  | 0.98 | 0.93 | 0.84 | 0.85 | 0.83 | 0.86 | 0.92 | 0.94 | 0.94 | 0.9 |

We added a sentence about the main limitations of SGRs (also see response to comment #4 from reviewer #2) in P15L13-14:

"*The performance of the method appears to be particularly related to the size of river reach.*"

We also changed sentences in the conclusion section (P19L3-5 and P19L9-11) to:

"*There were strong correlations between modelled storage and MODIS and GFDS water extent across the Amazon Basin. The Optimal Selection method is mainly limited by the size of river reach, as correlation generally increased from upstream to downstream as river width increased.*"

and

"*There were more potential SGRs derived using MODIS than GFDS, most likely because MODIS has higher spatial resolution than GFDS. However, GFDS SGRs predicted river discharges with more accuracy as GFDS was much less affected by cloud and dense vegetation than MODIS.*"

(2) The paper is quite well organized, but could be improved by: - explaining the concept of SGR in the introduction - providing a workflow scheme of the overall methodology - better justifying the interest and added value of SGR compared to existing methods and models.

Thank you for your suggestions. For explaining the concept of SGRs better, we changed the sentences in P3L17-23 to:

X

"*In this paper we investigate whether satellite gauging reaches (SGRs) can be established at both gauged and ungauged rivers and applied to provide continuous, consistent, and up-to-date river discharge monitoring over a large area. An SGR, by analogue of an in situ gauging station, is constructed based on an automated statistical method which relates hydrological model simulated river discharge to optical or passive microwave-derived surface water extent fraction for a region that includes the river reach.*"

We also added a figure with a workflow of the overall methodology in Figure 1 in the P4L7-9.

[Figure]

*Figure 1 Workflow of the overall methodology (rectangle: data; diamond: method; parallelogram: validation).*

We added discussion of our method compared to existing methods and models in P17L7-25:

"*Previous research demonstrated that both gauging data and hydrology modelling can be used to calibrate the remote sensing signal for estimating river discharge (Brakenridge et al., 2012; Revilla-Romero et al., 2014). Van Dijk et al. (2016) developed gauge-based SGRs using optical and passive microwave derived water extent observations, which is valuable to gap-fill and extend gauging discharge records. In addition to that, we demonstrated that SGRs can be also developed using hydrological modelling. We compared our model-based SGRs to gauge-based SGRs from previous research (Van Dijk et al., 2016) for all gauging reaches except gauging station G27 due to its unreliable records (Table 3). Both gauge-based and model-based GFDS SGRs at gauging station*

*G12 and G19 have higher Pearson correlations than the model, which suggests opportunities for data assimilation to improve the model. At gauging station G1, G5, G21, and G24, the model performs much better than both gauge-based and model-based SGRs, which suggests that uncertainties in SGRs at these locations mainly arise from remote sensing, e.g. due to cloud or dense vegetation. Errors and uncertainties of the model, such as from input data, routing, and conceptual structure, can also affect the performance of SGRs. For instance, for GFDS SGRs at gauging station G6 and MODIS SGRs at gauging station G31, gauge-based SGRs produced higher Pearson correlations than model-based SGRs. Compared to gauge-based SGRs, the main advantage of our method is the practical applicability in both gauged and ungauged rivers. Our results show that the model outperforms SGRs in most cases. Nonetheless, we consider SGR as an alternative, simple and automated approach for river discharge prediction using satellite observation only. For example, SGRs would be useful as an alternative if the model was unable to provide real-time estimations due to delayed rainfall estimates. As we used a model to train SGRs, poor model simulations might reduce the performance of SGRs. If more accurate and reliable hydrological models are available, SGRs can be redeveloped to estimate river discharge with greater accuracy.*"

(3) P6L3-4. Is there any quantitative criterion which motivated this choice?

$10^2$ m$^3$s$^{-1}$ was an empirical threshold that we chose as it shows the main river network in the Amazon Basin (Figure 2). We assume that rivers with mean discharge above $10^2$ m$^3$s$^{-1}$ have wider channels and broader floodplains than those with mean discharge below this threshold. In addition, annual mean discharge at all gauging stations ranges from 235 to 172,167 m$^3$s$^{-1}$ except G29 with very low discharge of 84 m$^3$s$^{-1}$. Thus, this threshold seemed a reasonable choice to cover river reaches with gauging stations for validation and evaluation.

(4) P6L12-13. Not clear

Agreed. We modified this sentence in P7L17-19 to:

"*For method B, spatial average water extent for the whole period of 2000-2014 was compared to storage directly, as this produces the same results as using the cross-validation method.*"

(5) P7L30-31. Is it a quantitative or qualitative result?

The Optimal Selection method was chosen based on quantitative analysis as it shows highest correlations compared to the Window Mean method, while the selection of window size was decided based on qualitative interpretation.

(6) Figure 4. Colors of SGR sites (purple and black) are not clearly visible.

Thank you for your suggestion. We improved this figure in P12 (Figure 5).

(7) Figure 5. The figure is too small.

Agreed. We enlarged it to cover two pages in P14-P15L3 (Figure 6).

We modified this sentence in P15L21 to:

> "*However, the performance of SGRs varies even for rivers of similar size.*"

Correct (see P16L12-22). To highlighting this point, we changed the sentence in P16L16-17 to:

> "*Here the SGRs are mainly based on the model, so we would expect the developed SGRs should have the ability to reflect flow patterns rather than absolute flow values because of model biases.*"

[Figure]

***Figure 5*** *Spearman correlation (ρ) between modelled river channel storage and MODIS (a) and GFDS (b) water extent using the Optimal Grid Cell Selection method (Method A) with a search window of 0.55 °×0.55 °(circle: gauging station; circle with black label: potential SGRs sites where gauging data is available).*

XIV

**Reviewer #4 Comments:**

In my opinion, this research topic is very important, and the authors have made great efforts in improving the technique of discharge estimation by combining the outputs from model and satellite products. However, the presentation and organization of the paper is very weak (although good grammar), make the readers very difficult to catch the key points and major conclusions. When you write a paper, you should consider the major conclusion/outputs, and the materials/work that can support the conclusion, making your "take-home-message" very clear; while not presenting all the subtle details in the manuscript. In addition, this paper only gives 5 figures, I would like to suggest adding quite a few tables to better summarize the contents (including your method, datasets, evaluation procedure, and so on).

(1) suggest rewriting the abstract, to make it more clear by only summarizing the major results and key conclusions. Your methodology should be described with better clarity.

We thank the reviewer for this constructive suggestion, and rewrote the abstract as follows:

> *"River discharge measurements have proven invaluable to monitor the global water cycle, assess flood risk, and guide water resource management. However, there is a delay, and ongoing decline, in the availability of gauging data, and stations are highly unevenly distributed globally. While not a substitute for river discharge measurement, remote sensing is a cost-effective technology to acquire information on river dynamics in situations where ground-based measurements are unavailable. The general approach has been to relate satellite observation to discharge measured in situ, which prevents its use for ungauged rivers. Alternatively, hydrological models are now available that can be used to estimate river discharge globally. While subject to greater errors and biases than measurements, model estimates of river discharge do expand the options for applying satellite-based discharge monitoring in ungauged rivers. Our aim was to test whether satellite gauging reaches (SGRs) can be constructed based on MODIS optical or GFDS passive microwave derived surface water extent fraction and simulated discharge from the World-Wide Water (W3) model version 2. We designed and tested two methods to develop SGRs across the Amazon Basin and found that the Optimal Grid Cell Selection method performed best for relating MODIS and GFDS water extent to simulated discharge. The number of potential river reaches to develop SGRs increases from upstream to downstream as rivers widen. MODIS SGRs are feasible for more river reaches than GFDS SGRs due to its higher spatial resolution. However, where they could be constructed, GFDS SGRs predicted discharge more accurately as observations were less affected by cloud and vegetation. We conclude that SGRs are suitable for automated large-scale application and offer a possibility to predict river discharge variations from satellite observations alone, for both gauged and ungauged rivers."*

We also added a workflow to explain our methodology better (see response to comment #2 from reviewer #3) in P4L7-9 (Figure 1).

(2) suggest removing much details in experiment setting, or use tables to list different methods/scenarios, and your calibration/validation periods.

Thank you for these suggestions. We used two tables to help describe our experiments (Table 1 and 2) in P6L28 and P7L21, respectively.

[revised manuscript text omitted]

---

## Author Response (AR2)

**Response to Reviewers**

**''Using modelled discharge to develop satellite-based river gauging: a case study for the Amazon Basin'' by Jiawei Hou et al.**

We thank the two reviewers for their constructive comments, which enabled us to improve the quality of our manuscript further. The reviewers' comments are reproduced below in blue, and our responses are presented in black.

**Reviewer #1 Comments:**

(1) As stated in my previous comment, my main concern is the approach limitation tied to the requirement of strong correlations between model outputs and satellite results. The authors argued that the approach will "be useful as an alternative if the model was unable to provide real-time estimations due to delayed rainfall estimates". Considering the GPM 3-hour to daily rainfall products, the "delayed rainfall estimates" problem seems to me solvable. Please clarify.

We acknowledged the approach limitation tied to the requirement of strong correlations between model outputs and satellite results and suggested ways to solve this limitation in P15L22-24:

> *"As we used a model to train SGRs, poor model simulations might reduce the performance of SGRs. If more accurate and reliable hydrological models are available, SGRs can be redeveloped to estimate river discharge with greater accuracy."*

and for clarifying the usefulness of SGRs, we changed the sentence in P15L20-22 to:

> *"SGRs would be useful as an alternative if the model was unable to provide real-time estimates, e.g. due to delayed rainfall gauge observations."*

(2) P15L25 to P16L1: "…MODIS optical remote sensing is limited to clear-sky conditions and requires surface water to be unobscured by a dense vegetation canopy, whereas GFDS passive microwave remote sensing is much less affected by either of these factors". It is correct to say GFDS or 36.5 GHz microwave is less affected by atmospheric conditions, but dense vegetation can still block most 36.5 GHz microwave signals of underlying soil/water. Please re-phrase.

Agreed. We modified the sentence in P13L24-P14L1 to:

> *"A likely reason for this is that MODIS optical remote sensing is limited to clear-sky conditions, whereas GFDS passive microwave remote sensing is much less affected by this."*

**Reviewer #3 Comments:**

(1) In my opinion, the authors clearly answered all the remarks I raised, except the addition of a short explanation of the SGR concept. The authors added a sentence to describe how it is constructed, but not what it is. van Dijk et al. (2016), who introduced this concept, defined SGRs as "remote sensing equivalent of in situ gauging stations". In contrast, in the community of satellite altimetry for hydrology, this concept is referred as "virtual station". I can understand that considering a reach instead of a station is more appropriate when analysing surface water extent. Maybe the authors could just add one or two sentences to clearly state this (and a few words in the abstract as well). Apart from that, I think the manuscript is ready for publication.

Thank you for your suggestion. For explaining the concept of SGRs better, we added a sentence in P3L11-13:

[revised manuscript text omitted]